

# Beyond HCHO/NO₂: Global Daily Maps of Net Ozone Production Rates and Sensitivities Constrained by Satellite Observations (2005–2023)

**Amir H. Souri[1,2]\*, Gonzalo González Abad[3], Bryan N. Duncan[1], Luke D. Oman[1]**

[1]Atmospheric Chemistry and Dynamics Laboratory, NASA Goddard Space Flight Center, Greenbelt, MD, USA

[2]GESTAR II, Morgan State University, Baltimore, MD, USA

[3]Atomic and Molecular Physics (AMP) Division, Center for Astrophysics | Harvard & Smithsonian, Cambridge, MA, USA

\* Corresponding author: a.souri@nasa.gov

**Abstract.** Previous studies on net ozone production rates (PO₃) and their sensitivities to precursors relied on limited in-situ data, often coarse and uncertain chemical transport models (CTMs), and ozone indicators like the formaldehyde-to-nitrogen dioxide ratio (FNR). However, FNR fails to fully capture PO₃'s complex relationships with pollution, light, and water vapor. To address this, we refine the satellite-based PO₃ product from Souri et al. (2025) with key advancements: (i) a deep neural network to parametrize high-dimensional non-linear ozone chemistry without the need for empirical linearization of atmospheric conditions, (ii) incorporation of water vapor, (iii) improved error characterization, and (iv) the application of a finer CTM to dynamically convert column retrievals into near-surface mixing ratios. Our PO₃ sensitivity maps surpass traditional FNR-based assessments by quantifying sensitivity magnitudes – factoring in photolysis rates and water vapor – with greater spatial information. Our PO₃ product with its high horizontal coverage will advance our understanding of the drivers of locally-produced ozone pollution, but only at a single snapshot per day. Specifically, our new product provides daily near-clear sky PO₃ and sensitivity maps using bias-corrected OMI (2005-2019, 0.25° × 0.25°) and TROPOMI (2018-2023, 0.1° × 0.1°), with values aligning within 10%. High PO₃ rates (>8 ppbv/hr) appear in urban and biomass-burning regions under strong photochemical activity, including during a heatwave in the northeastern U.S. Photolysis rates are the dominant factor dictating the seasonality of PO₃ magnitudes and sensitivities.

## 1. Introduction.

To mitigate  tropospheric ozone pollution, a pervasive trace gas that impacts human health, climate, and crop productivity (Fleming et al., 2018; Mills et al., 2018; Gaudel et al., 2018), it is essential to quantify the spatiotemporal variations of two primary components: i) the sensitivity of the chemical net production rates of ozone (PO₃) to its two main precursors, nitrogen oxides (NOₓ=NO+NO₂) and volatile organic compounds (VOCs), and ii) the magnitude of PO₃ itself. The first component provides insights into the positive and negative contributions of these precursors to PO₃, which are typically categorized as NOₓ-sensitive (where PO₃ is influenced mainly by NOₓ), VOC-sensitive (where PO₃ is affected primarily by VOCs), and transitional regimes (where PO₃ is responsive to both NOₓ and VOCs) (Kleinman et al., 2002; Silman and He, 2002; Duncan et al., 2010). The latter component is crucial for understanding how locally produced ozone, in conjunction with advected or diffused ozone, can lead to high-ozone events (e.g., Kleinman et al., 2002, 2005; Sullivan et al., 2019).

Creating global maps of PO₃ and its sensitivity at spatiotemporal scales relevant to air quality policies is a challenge. Unique instruments can directly measure PO₃ by calculating the difference in ozone molecules from air samples drawn through two distinct tubes – one exposed to sunlight and the other



shielded by an ultra-violet (UV) filter (Cazorla and Brune, 2010; Sadanaga et al., 2017; Sklaveniti et al.,
2018). However, these instruments suffer from various interferences, such as heterogeneous chemistry or
photo-enhanced loss of ozone within the tubes, and they are limited to sparse super sites that restrict spatial
variability. Similarly, box-model simulations of $PO_3$, which are observationally constrained by intensive
atmospheric composition measurements, are also limited by sparse aircraft sampling (Cazorla et al., 2012;
Ren et al., 2013; Mazzuca et al.; 2016; Souri et al., 2020a; Schroeder et al., 2020; Brune et al., 2022; Wolfe
et al., 2022; Souri et al., 2023a). Currently, our understanding of the global spatiotemporal variability of
$PO_3$ mainly relies on chemical transport models, which can possess significant uncertainties such as those
associated with transport, emissions, and dry deposition. Moreover, they may lack the spatial resolution
necessary to capture the non-linear dynamics associated with $NO_X$ and thus, ozone chemistry (Valin et al.,
2011; Vinken et al., 2011; Yu et al., 2016).

The "gold standard" approach to determine three-dimensional $PO_3$ within a process-based
framework involves running a high-resolution chemical transport model, with prognostic inputs constrained
by observations. This approach falls into the realm of an inversion/data assimilation framework (Bocquet
et al., 2015). Numerous studies have aimed to constrain various model prognostic inputs, including $NO_X$
and VOCs emissions and/or concentrations, using aircraft and satellite remote sensing retrievals (e.g.,
Stavrakou et al., 2009, 2016; Souri et al., 2016; Bauwens et al., 2016; Miyazaki et al., 2020; Opacka et al.,
2025). Notably, Souri et al. (2020b) developed a non-linear joint inversion of $NO_X$ and VOCs to better
constrain $PO_3$, thereby shedding light on the impact of recent emission regulations in East Asia on the
different chemical pathways governing the formation and loss of surface ozone. However, these studies
face a fundamental challenge: discrepancies between simulated fields and observations are often blamed
solely on emissions. In fact, such discrepancies can also stem from various model components, including
chemical mechanisms, dry deposition, photolysis rates, vertical diffusion, and transport. Given the limited
observations available for constraining all of these uncertain parameters, the optimization problem becomes
grossly under-determined. This means we lack sufficient information to uniquely determine the optimal
values of these parameters altogether. Additionally, the underlying physics of these models is inherently
uncertain, necessitating the explicit propagation of model physics errors into our final estimates or the
execution of ensemble model realizations to vet the credibility of the top-down estimates across different
realizations from a stochastic point of view. Conducting these ensemble optimizations at fine-scale grid
boxes around the globe is prohibitively computationally intensive.

At the expense of sacrificing the full capability of a physics-based model, we can take advantage
of a statistical approach to predict $PO_3$ using several observable variables with improved computational
efficiency. Chatfield et al. (2010) made an early effort to parameterize the gross production of ozone via
$NO+HO_2$ through a multivariable power law function that depended on formaldehyde (HCHO), nitrogen
dioxide ($NO_2$), UV photolysis rates, and ambient temperature. Their model successfully reproduced over
60% of the variance observed in the ozone gross production rates. Souri et al. (2023a) introduced a bilinear
equation based on $HCHO \times NO_2$ and $HCHO/NO_2$, which explained more than 80% of the variance in
simulated $PO_3$. Building on these findings, Souri et al. (2025) developed a regularized piece-wise linear
regression to parameterize $PO_3$ using retrospective aircraft observations and a set of variables, including
$HCHO/NO_2$, HCHO, $NO_2$, $jO^1D$ (photolysis frequency for $O^1D+h\nu$), and $jNO_2$ (photolysis frequency for
$NO_2+h\nu$). Their algorithm successfully reproduced over 90% of the variance in observationally-constrained
$PO_3$ with minimal biases across moderately to extremely polluted regions.

These parameterizations present a unique opportunity to globally map $PO_3$, as their primary inputs
can be largely constrained by well-characterized satellite retrievals with extensive horizontal coverage
(Gonzalez Abad et al. 2019). For this reason, Souri et al. (2025), compiled various satellite observations
including TROPOspheric Monitoring Instrument (TROPOMI) surface albedo, HCHO, and $NO_2$ columns
in conjunction with pre-computed model fields to populate the inputs to their parametrization, allowing
them to generate the first-ever maps of $PO_3$ worldwide. Because their algorithm had an explicit
mathematical form, they were also able to break down $PO_3$ into HCHO and $NO_2$ contributions, providing





much more detailed spatial information about ozone sensitivity maps compared to binary information (i.e., $NO_X$-sensitive or VOC-sensitive) made from HCHO to $NO_2$ ratios (Martin et al., 2004; Duncan et al., 2010; Choi et al., 2012; Choi and Souri, 2015a, b; Jin et al., 2017; Schroeder et al., 2017; Souri et al., 2017; Jeon et al., 2018; Tao et al., 2022; Jonhson et al., 2024). However, HCHO to $NO_2$ ratios (known as formaldehyde to nitrogen dioxide ratios – FNR) were a central component of their algorithm to transform the non-linear ozone chemistry into several linear segments (i.e., a piecewise regression).

The inclusion of FNR in Souri et al. (2025) might introduce several complications, such as i) the amplification of unresolved systematic and random errors in satellite retrievals associated with $PO_3$ estimates, and ii) discounting the dependency of $PO_3$ sensitivity to HCHO and $NO_2$ concentrations as function of available light and water vapor. In fact, FNR does not provide useful information about ozone chemistry in less photochemically active environments, such as early morning or late afternoon conditions (known as light-limited or radical-limited conditions). Although the parametrization of $PO_3$ crafted in Souri et al. (2025) relied on photolysis rates, the sensitivity of $PO_3$ to $NO_2$ (a proxy for reactive nitrogen) and HCHO (a proxy for VOC reactivity) did not directly depend on photolysis rates. Therefore, the present work aims to enhance the capability of the algorithm designed in Souri et al. (2025) using a machine learning approach that can effectively establish a robust non-linear relationship between $PO_3$ and various observable geophysical variables without the need for segregation or linearization based on concentrations of ozone precursors, light intensity, or humidity.

The new product of $PO_3$ along with spatially varying ozone sensitivity maps using bias-corrected OMI and TROPOMI retrievals are generated globally for 2005-2023. We will document the advantages of this algorithm over the older one and how the new results can bring fresh insights into $PO_3$ behavior across various seasons, locations, and anomalous events (e.g., heatwaves).

## 2. Data

### 2.1. Satellite Retrievals

#### 2.1.1. TROPOMI HCHO and $NO_2$

We use daily level-2 (L2) products of TROPOMI (v2.4-v2.5) tropospheric $NO_2$ and total HCHO columns (v2.4-v2.6) obtained from UV-Vis radiances (~328-496 nm) onboard the European Space Agency's (ESA's) Sentinel Precursor (S5P) spacecraft with an equatorial overpass time of ~1330 local standard time (LST) (Veefkind et al., 2012; van Geffen et al. 2022; De Smedt et al. 2021). These products offer near-daily global coverage of $NO_2$ and HCHO columns at a horizontal resolution of 7.2 km (reduced to 5.6 km after August 2019) by 3.6 km at nadir, extending to approximately 14 km at the edges of the scanline, with a swath width of 2600 km. The data products used in this study span from May 2018 to the end of 2024. The retrieval process follows a two-step framework: first, a differential spectral fitting algorithm is used to determine the number of integrated molecules along the slant light path, and second, air mass factor calculations are done based on simulated gas absorber profiles and radiative transfer model calculations to convert slant columns into vertical ones.

Both products have been thoroughly vetted against ground-based remote sensing retrievals, including the multi-axis differential optical absorption spectrometer (MAX-DOAS) (De Smedt et al., 2021; Verhoelst et al., 2021; van Geffen; Souri et al., 2025) and Fourier transform infrared spectroscopy (FTIR) (Vigouroux et al., 2020; Souri et al., 2025), showing a general tendency towards underestimation in polluted regions. We include in our study only pixels with a quality flag (*q_value*) exceeding 0.5 and 0.75 for HCHO and $NO_2$ products, respectively. The quality flag encapsulates errors coming from clouds, snow, surface refractivity, and algorithm performance. The selected values are based on the user manual recommendation (Eskes et al., 2020; De Smedt et al., 2021). Additionally, we bias correct the columns using the offset (additive term) and slope (multiplicative term) determined from a global comparison of MAX-DOAS/FTIR



and these datasets, as described by Souri et al. (2025). The resulting daily bias-corrected HCHO and $NO_2$
columns, along with the retrieval errors, are mapped onto a 0.1°×0.1° global grid using a mass-conserved
bilinear interpolation approach described in Souri et al. (2024).

### 2.1.2.  OMI HCHO and $NO_2$

We use the Quality Assurance for the Essential Climate Variables (QA4ECV) $NO_2$ daily Level 2
product (Boersma et al., 2018) which is based on global radiances captured by the Ozone Monitoring
Instrument (OMI) sensor aboard NASA's Aura spacecraft. This product is retrieved with a similar overpass
time as TROPOMI. The horizontal resolution of the product ranges from 13×24 km² at nadir to 165×13
km² at the edge of the scanline. It relies on OMI Collection 3 radiance data. Since 2008, OMI has faced
significant anomalies resulting in the loss of reliable data in areas of its detectors, a situation referred to as
the "row anomaly." This has led to inconsistent spatial resolution and global coverage throughout its
operational phase. However, the unaffected detectors have demonstrated a high level of stability over the
past two decades, making this product suitable for long-term trend analysis. Detailed description of the
retrieval algorithm, along with validation against ground remote sensing data, can be found in Boersma et
al. (2018), Compernolle et al. (2020), and Pinardi et al. (2020). We include good quality pixels based on an
effective cloud fraction below 50%, a quality processing flag parameter equal to zero, and exclusion of
snowy regions. Additionally, we discard the last two rows of the detector because of their poor horizontal
resolution. We use the OMI $NO_2$ product for the period from 2005 until the end of 2019. To correct for
offset and slope in this product, we apply the statistics from Pinardi et al. (2020), who compared this product
against MAX-DOAS observations, factoring in an empirical dilution adjustment to address the mismatch
between the OMI resolution and MAX-DOAS.
We also use the OMI Smithsonian astrophysical observatory (SAO) daily HCHO Level 2 product
from the same sensor, which is generated using a newly developed algorithm and Collection 4 OMI
radiances (Ayazpour et al. 2025; Nowlan et al., 2023). This improved algorithm enhances the radiance
information content used to retrieved HCHO columns, significantly reducing noise in the slant column fit.
The stability of this product in extracting new information related to long-term global trends of HCHO has
been well demonstrated in recent studies (Souri et al., 2024; Anderson et al., 2024). Ayazpour et al. (2025)
validated the product against the global FTIR network and found that the HCHO columns tend to be biased
low in polluted regions. Accordingly, we use their statistics to adjust for both offset and slope associated
with the data. We include only good data following the quality flag provided with the dataset along with
effective cloud fraction below 40%. Both bias-corrected OMI products are mapped onto a global grid with
a resolution of 0.25°×0.25° using the same algorithm used for TROPOMI.

### 2.1.3.  Surface albedo

To estimate near-surface photolysis rates of $jO^1D$ ($O_3$+hv, <350 nm) and $jNO_2$ ($NO_2$+hv, ~400-500
nm) used in the parametrization of $PO_3$, we are required to provide reasonable surface albedo estimates
(Section 2.4). We use a monthly Directionally Dependent Lambertian-Equivalent reflectivity (DLER)
climatology derived from TROPOMI radiances at the spatial resolution of 0.125°×0.125°; the product is in
good agreement with the MODIS BRDF product (Tilstra et al., 2024). This climatology has two sets of
values for both shortwave (328 nm) and longwave UV (463 nm) that are used separately for calculating
$jO^1D$ and $jNO_2$, respectively. We use only the isotropic part of the DLER product (named *minimum_LER*),
which is added to an offset coefficient provided with the dataset.

### 2.2.  *Aircraft Measurements*

The use of aircraft observations is twofold: first, they provide a vast number of measured
geophysical variables suitable to simulate our observationally-constrained $PO_3$ training dataset (Section
3.1); second, they enable a rigorous validation of column-to-PBL conversion factors derived from a
chemical transport model (Appendix B). We use the dataset compiled by Souri et al. (2025), who curated





various aircraft campaigns measuring photolysis rates, meteorological variables, and atmospheric
composition from varying atmospheric conditions, including urban/suburban settings (DISCOVER-AQs,
and KORUS-AQ), high-vegetated regions (SENEX), and remote areas (INTEX-B and AToms). The
sampling frequency varies from 10-sec to 30-sec. More detailed information regarding the choice of
instrument, gap filling, and data exclusion can be found in Souri et al. (2025).

### 2.3. MINDS simulations

We use a global chemical transport model simulation designed to support trace gas retrievals. The
simulation, called Multi-Decadal Nitrogen Dioxide and Derived Products from Satellites (MINDS), was
generated using the Goddard Earth Observing System (GEOS) Earth system model (Molod et al., 2015;
Nielsen et al., 2017) equipped with the full chemistry Global Modeling Initiative (GMI) mechanism
(Duncan et al., 2007; Strahan et al., 2007) and coupled with the Goddard Chemistry Aerosol Radiation and
Transport (GOCART) aerosol module (Chin et al., 2002). The rapid radiative transfer model, which was
designed for global climate models (GCMs) and is known as the Radiative Transfer Module for GCM
(RRTMG), calculates the longwave and shortwave radiation influenced by aerosols simulated by GOCART,
enabling the incorporation of the direct effects of aerosols on meteorological conditions (Nielsen et al.,
2017). The model is setup at c360 grid (0.25°×0.25°) and covers the period of 1993 until the end of 2023.
The model follows 72 hybrid sigma values ranging from the surface to 0.01 hPa. Several prognostics inputs
related to meteorology, including water vapor, are constrained by MERRA-2 reanalysis (Orbe et al., 2017).
Lightning production of NO is parametrized based on the simulated convection. The model uses the
Monitoring Atmospheric Chemistry and Climate and CityZen (MACCity) inventory (Granier et al., 2011)
of anthropogenic emissions downscaled to 0.1°×0.1° using the Emissions Database for Global Atmospheric
Research version 4.2 (EDGAR 4.2). These anthropogenic emissions change by year and month. Biomass
burning emissions rely on the Fire Energetics and Emissions Research (FEER) dataset (Ichoku and Ellison,
2014). Biogenic emissions are modeled by the Model of Emissions of Gases and Aerosols from Nature
(MEGAN) v2.1 (Guenther et al. 2012). It is known that isoprene emissions in MEGANv2.1 are largely
overestimated (Bauwens et al., 2016; Souri et al., 2020b), therefore they are scaled down by a factor of two.

### 2.4. TUV NCAR Photolysis Rates Look-up Table

To estimate $jNO_2$ and $jO^1D$, we refer to a detailed look-up table provided by the Framework for 0-
D Atmospheric Modeling (F0AM) model (Wolfe et al. 2016). This table is developed for clear-sky
conditions based on over 20,064 solar spectra calculations. The data encompasses a broad spectrum of solar
zenith angles (SZA) from 0° to 90° in 5° increments, altitudes ranging from 0 to 15 km in 1 km steps,
overhead total ozone columns from 100 to 600 DU in increments of 50 DU, and surface UV albedo values
from 0 to 1 in 0.2 increments. These calculations were carried out using NCAR's Tropospheric Ultraviolet
and Visible radiation model (TUV v5.2), along with cross sections and quantum yields from IUPAC and
JPL (Wolfe et al., 2016). Information on SZA and surface elevation is obtained from the L2 TROPOMI/OMI
granule data. Surface albedo is based on the TROPOMI DLER climatology (Section 2.1.3). The overhead
total ozone columns are derived from MINDS simulations (Section 2.3). For any values that fall between
the entries in the tables, we apply a linear interpolation method.

### 2.5. Empirical $PO_3$ estimates using LASSO

We will compare our new product (Section 3.2) to an empirical method developed by Souri et al.
(2025), who took advantage of simulated $PO_3$ data constrained by aircraft measurements to parameterize
$PO_3$ using four geophysical variables: $NO_2$, HCHO, $jNO_2$, and $jO^1D$. Their algorithm used a piecewise L1-
regularized linear regression model known as Least Absolute Shrinkage and Selection Operator (LASSO).
Since the algorithm was based on a linear model which was ill-suited for the non-linear ozone chemistry, it
was necessary to linearize the parameterization using various thresholds for FNRs. Despite the method's
simplicity, Souri et al. (2025) were able to reproduce approximately 88% of the variance with low biases
(less than 20%) in observationally-constrained $PO_3$. Using the empirical method, they generated the first





230 maps of $PO_3$ by combining bias-corrected TROPOMI HCHO and $NO_2$ columns, simulated photolysis rates,
231 and a global transport model designed for the conversion from column measurements to the planetary
232 boundary layer (PBL).

233 To isolate the performance of the $PO_3$ estimator used in Souri et al. (2025) as compared to the
234 proposed algorithm in this study, we will ensure that the input variables, including the mixing ratios of
235 HCHO and $NO_2$ within the PBL as well as the photolysis rates, remain identical for both the empirical
236 product and our new algorithm. Henceforth, we will refer to this empirical product as "$PO_3$LASSO".

### 237 3. **Methodology**

238 Figure 1 illustrates the three-stage process of our newly developed algorithm to operationally
239 produce long-term maps of $PO_3$ within the PBL along with the sensitivity and error maps. The product is
240 called "$PO_3$DNN".

241 *Stage I* –This stage serves as the foundation for the product, focusing on parameterizing $PO_3$ using
242 a regularized Deep Neural Network (DNN). The training dataset, detailed in Section 3.1, is derived from
243 an observationally-constrained F0AM box model that provides simulated $PO_3$ along with various
244 atmospheric quantities directly or indirectly constrained by aircraft measurements. The decision to make
245 use of aircraft data is based on two main factors: i) they capture real-world atmospheric conditions across
246 diverse parts of the atmosphere and various geographic regions, and ii) the significant fluctuations inherent
247 in the data rigorously test the DNN's capability to generalize (i.e., to fit the model through the data rather
248 than merely to the data). However, a notable limitation of aircraft data is its restriction to specific
249 atmospheric conditions. To address this, we have expanded the training dataset by perturbing the inputs to
250 the F0AM model (Section 3.1), resulting in a synthetic dataset. This expanded training dataset is then used
251 for validation, testing, and calibration of the DNN algorithm.

252 *Stage II* – The objective of this stage is to prepare spatiotemporal geophysical variables necessary
253 for the prediction of $PO_3$ (done in Stage III). We need five parameters on a global scale with daily frequency:
254 $jNO_2$, $jO^1D$, HCHO, $NO_2$, and $H_2O(v)$. To generate global daily maps of near-surface photolysis rates, we
255 use the NCAR's look-up table as detailed in Section 2.4; this table relies on SZA, which varies with time
256 and location, as well as surface UV-Vis albedo, ozone overhead columns, and surface altitudes. Data on
257 both SZA and surface altitude are acquired from the satellite L2 products. Ozone overhead columns are
258 from MINDS. For surface UV-Vis albedo, we use two different wavelengths based on TROPOMI's
259 climatology (Section 2.1.3). These calculations assume clear sky conditions, which are somewhat achieved
260 by the effective cloud fraction thresholds derived from both the OMI and TROPOMI products. However,
261 the presence of partially cloudy pixels and aerosols can introduce uncertainties in calculated photolysis
262 rates. To address these effects using a radiative transfer model (e.g., RRTMG), long-term observations of
263 three-dimensional optical properties of clouds and aerosols on a global scale are needed. Unfortunately,
264 such records are typically limited to narrow paths of spaceborne lidar observations, which carry
265 considerable uncertainties (Thorsen and Fu, 2015). Assuming that both clouds and aerosols attenuate UV-
266 Vis sunlight down to the surface—creating a shielding effect rather than a brightening effect—over bright
267 urban areas, the omission of aerosols and clouds in our product is likely to result in an overestimation of
268 near-surface photolysis rates over those areas. Our algorithm uses HCHO and $NO_2$ columns obtained from
269 OMI or TROPOMI, which are bias-corrected against ground remote sensing data. These measurements are
270 then transformed into the mixing ratios in the PBL region using the vertical distribution of HCHO and $NO_2$
271 profiles simulated by MINDS. The final variable is the average number of water vapor ($H_2O(v)$) molecules
272 per cubic meters in the PBL region at the satellite overpass time, which is obtained directly from the MINDS
273 simulation. It is important to note that the MINDS simulation is based on constraints from MERRA-2
274 reanalysis, underscoring that the $H_2O(v)$ simulations are constrained by observations.



*Stage III* – In the final stage, we predict $PO_3$, generate sensitivity maps, and provide both systematic
and random errors associated with these estimates. To create $PO_3$ maps, we input the five parameters from
Stage II into the DNN model developed in Stage I. To generate the sensitivity maps of $PO_3$ in relation to
$NO_2$ and HCHO, we apply perturbations to $NO_2$ and HCHO based on the methodology described in Section
3.3. These perturbations also serve another purpose which is to propagate the errors associated with the
retrievals of HCHO and $NO_2$, as well as their corresponding conversion factors from MINDS into the final
product. A comprehensive explanation of the error budget and characterization can be found in Section 3.4.

While we perform Stage I only once to establish a $PO_3$ estimator, we need to run Stage II and III
for any desired location/time or spatial resolution. The need to operationally run these two stages has
motivated us to create an open-source and object-oriented Python package called *ozonerates* v1.0 (Souri
and Gonzalez Abad, 2025), which is capable of running all steps while leveraging parallel computation.



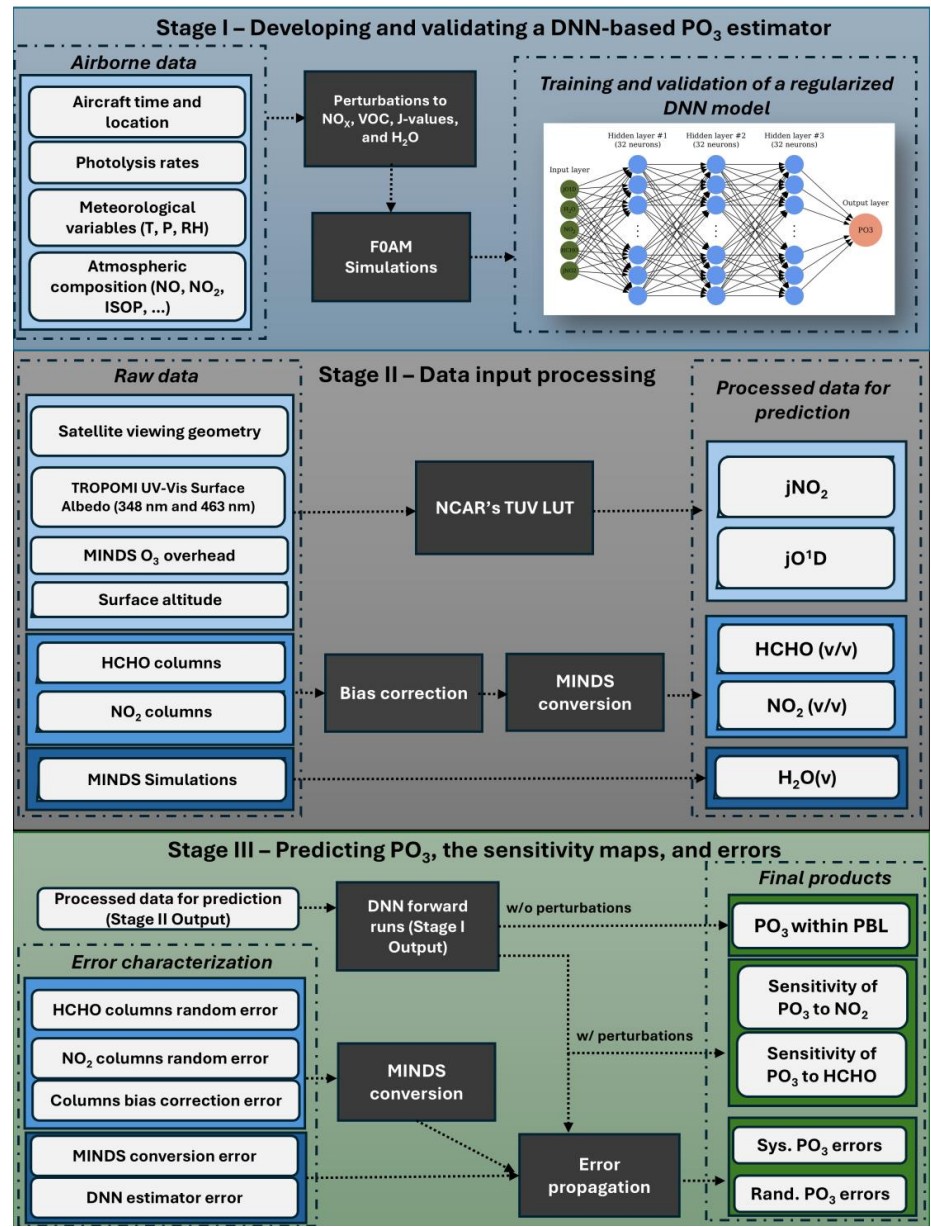

**Figure 1.** Processing stages developed to operationally generate PO$_3$ and sensitivity maps along with daily frequency errors on a global scale. Stage I aims to establish a regularized DNN model based on synthetic and real-world aircraft measurements. Stage II prepares the necessary satellite-based input features used for PO$_3$ prediction in Stage III. Stage III feeds the DNN model with Stage II values and some statistical error analysis to populate the final product.



*3.1.*    ***Training dataset generation using F0AM box model***

To establish a relationship between several geophysical variables related to $PO_3$, we use F0AM
version 4 box model (Wolfe et al., 2016). This model is capable of simulating detailed chemical kinetics
based on user inputs regarding meteorological variables, atmospheric compositions, and photolysis rates.
F0AM uses a solver for ordinary differential equations (ODEs) designed for stiff systems, which allows it
to determine the chemical evolution of all species included in the selected chemical mechanism. We adhere
to previous configurations that apply the Carbon Bond 6 (CB06, r2) chemical mechanism within F0AM
(Souri et al., 2020a; Souri et al., 2023a; Souri et al., 2025). The model is constrained by data collected
during aircraft campaigns, including meteorological data, photolysis rates, and various trace gas
concentrations. Additional details regarding the selection of instruments, bias corrections for photolysis,
choices of dilution factors, and other configurations can be found in Souri et al. (2025). We incorporate data
from seven aircraft campaigns, including DISCOVER-AQ (Texas, Washington, Colorado), KORUS-AQ,
ATOMs, INTEX-B, and SENEX, to further constrain the model. Souri et al. (2025) demonstrated that this
setup effectively reproduces several unconstrained yet measured compounds, such as HCHO, $HO_2$, OH,
and PAN; moreover, the performance of the model was on par with other studies (e.g., Brune et al., 2020;
Brune et al., 2022; Miller and Brune, 2022), indicating that it is a suitable model setup for understanding
local ozone chemistry. This model-derived dataset consists of ~134k points.
A limitation to the training dataset prepared by Souri et al. (2025) originates from the fact that only
a subset of atmospheric conditions could be observed by the suborbital missions. A remedy for this
limitation is to synthetically regenerate data by systematically perturbing several of the inputs used in the
F0AM model. As a result, we apply a scaling factor, ranging from 0.1 up to 10 in 12 evenly-spaced steps,
separately to $NO_X$, VOCs, $H_2O(v)$, and photolysis rates. This expands the dataset to ~6.4 million datapoints,
covering a much wider range of atmospheric states. Once the simulations are done, we determine simulated
$PO_3$ by:

$$PO_3 = FO_3 - LO_3 \tag{1}$$

where $LO_3$ is all possible chemical loss pathways of ozone (negative stoichiometric multiplier matrix) and
$FO_3$ is all possible chemical pathways producing ozone molecules (positive stoichiometric multiplier
matrix). This equation is also known as ozone tendency.
*3.2.*    ***DNN architecture and configuration***

The overall architecture of the DNN model is portrayed in Figure 2. The design consists of three
fully-connected hidden layers each having 32 neurons. The neurons are equipped with rectified linear unit
(ReLU) activation functions. The training dataset (~6.4 millions) is split into 20% test, 24% validation, and
56% training. Training inputs to the parametrization consists of HCHO, $NO_2$, $jO^1D$, $jNO_2$, and $H_2O(v)$.
Prior to the training, we normalize them, such that each feature ($x$) is rescaled according to $x' = \frac{x-\mu}{\sigma}x$,
where $\mu$ and $\sigma$ represent the mean and standard deviation of the feature, respectively, ensuring a mean of
zero and a variance of one. The optimization (training) of the DNN follows the backpropagation rule armed
with Adaptive Moment Estimation (ADAM) optimizer which is known to perform well with noisy data
(Kingma and Ba, 2014). The initial learning rate is set to $10^{-5}$. We use 500 epochs. The loss function ($L$) of
the optimalization problem is:

$$L = \frac{1}{2}\sum_{k=1}^{N}(y_k - o_k)^2 + \lambda\sum_{i=1}^{p} w_i^2 \tag{2}$$

where the first term on the right side represents the mean squares error (MSE) of the prediction derived
from difference between the target $PO_3$ ($y$) and the predicted $PO_3$ ($o$). $N$ represents the number of training




datapoints. The second term is L2-regularization with a factor of $\lambda$ to reduce the squares of $p$ number of
neuron weights ($w$).

An important aspect of this optimization is the use of L2 regularization, which effectively helped
us determine the optimal number of hidden layers and neurons. L2 regularization penalizes the cost function
if an illusion of high prediction accuracy (the first term) is achieved with excessive variance in the solution
(weights). Failing to balance the prediction error and the solution variance can lead to overfitting, which
harms model performance in two ways: i) it results in erroneous predictions for atmospheric conditions that
fall outside the training dataset; ii) it diminishes the physical interpretability of the statistical model because
of large fluctuations in the weights, a common issue in regression models known as collinearity. When we
used too many neurons or layers, the regularization penalized the weights, causing a substantial proportion
to approach zero (not shown), indicating that those neurons were unnecessary. However, incorporating
regularization does have some drawbacks: i) it requires a smaller initial learning rate (set to $10^{-5}$) to avoid
falling into local minima, which demands more computational resources; and ii) the regularization factor
also needs to be optimized. We found that a value of $\lambda = 10^{-5}$ provided the best results among the set of
values [$10^{-4}$, $10^{-5}$, and $10^{-6}$], based on the symmetry in the statistical distributions of the test residuals, MSE,
and the overall level of physical interpretability observed in the sensitivity tests.

The implementation of the DNN model is done using the open-source *TensorFlow* application
programming interface (API) package in *Python* (Abadi et al., 2016). To thoroughly validate the
performance of this model from various angles we i) compare the DNN prediction with the test data using
various standard metrics, ii) investigate the evolution of the loss function derived from both the training set
and the validation one over epochs, iii) study the physical explanation of the response of $PO_3$ to $NO_2$ and
HCHO, water vapor, and photolysis rates, and iv) finally compare the DNN results to $PO_3$LASSO. We will
use a number of statistical metrics, including the coefficient of the determination ($R^2$), mean bias, mean
square error, mean absolute error, and root mean square error (RMSE), to carry out the quantitative
assessment (Section 4.1).

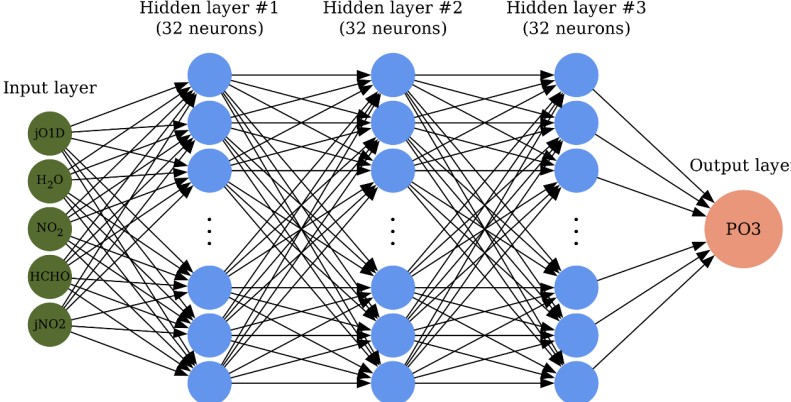


**Figure 2.** The architecture of the DNN model. The model contains three hidden layers with 32 neurons
each.
*3.3.  Sensitivity calculations*

To elucidate the response of $PO_3$ to its inputs, we calculate the semi-normalized sensitivities
through the finite difference method:

$$SPO3_i = \frac{[PO3]_i^{110\%} - [PO3]_i^{90\%}}{0.2} \qquad (3)$$



where $[PO3]_i^{110\%}$ and $[PO3]_i^{90\%}$ are PO$_3$ from perturbing input parameters ($i$=1 for NO$_2$, and $i$=2 for
HCHO) by 1.1 and 0.9 scaling factors. A mathematical proof showing that these sensitivity calculations
are equivalent to the directional derivative is provided in Appendix A.
*3.4.*   ***Error budget and characterization***
Since the PO$_3$DNN integrates atmospheric models, satellite trace gas retrievals, ground remote
sensing, and a machine learning approach, it contains various sources of errors, some of which will be
formulated in this section. Spatially and temporally averaging satellite-based products is a common practice
to reduce noise and fill gaps; therefore, we attempt to separate systematic errors (irreducible by averaging)
from random ones (reducible by averaging). We assign the total PO$_3$ within PBL region error ($e_{total}$) based
on the following equation:

$$e_{total} = \sqrt{e_{syst}^2 + e_{rand}^2} \qquad (4)$$

where $e_{syst}$ and $e_{rand}$ are systematic and random errors associated with PO$_3$ estimates. Systematic errors
account for the errors associated with the bias correction of OMI and TROPOMI against ground remote
sensing retrievals ($e_{HCHO\_bias\_c}$ and $e_{NO2\_bias\_c}$), the model-based conversion of columns to the PBL mixing
ratios ($e_{HCHO\_conversion}$, $e_{NO2\_conversion}$), and the DNN estimator error ($e_{DDN}$), and are given by:
$e_{syst} = \sqrt{e_{HCHO\_bias\_c}^2 + e_{NO2\_bias\_c}^2 + e_{HCHO\_conversion}^2 + e_{NO2\_conversion}^2 + e_{DNN}^2}$ \qquad (5)
$e_{HCHO\_bias\_c}^2 = \left( \frac{\partial PO_3}{\partial HCHO} \cdot \gamma \cdot e_{bc-HCHO} \right)^2$ \qquad (6)
$e_{NO2\_bias\_c}^2 = \left( \frac{\partial PO_3}{\partial NO_2} \cdot \gamma \cdot e_{bc-NO_2} \right)^2$ \qquad (7)
$e_{HCHO\_conversion}^2 = \left( \frac{\partial PO_3}{\partial HCHO} \cdot VCD_{HCHO} \cdot e_{conv-HCHO} \right)^2$ \qquad (8)
$e_{NO2\_conversion}^2 = \left( \frac{\partial PO_3}{\partial NO_2} \cdot VCD_{NO_2} \cdot e_{conv-NO_2} \right)^2$ \qquad (9)
where $\gamma$ is the conversion factor of the satellite total to the PBL columns translation based on MINDS and
the formulation by Souri et al. (2025); $e_{bc-HCHO}$ and $e_{bc-NO2}$, in column units, are calculated following the
formulation from Souri et al. (2025) who used the errors of slope and offset obtained from the comparison
of satellite VCDs to ground remote sensing benchmarks; $e_{conv-HCHO}$ and $e_{conv-NO2}$ are quantified by validating
the simulated conversion factors compared to those of aircraft vertical spirals (Appendix B). The unit for
these two errors is ppbv per the column unit; accordingly, we multiply these terms to satellite VCDs. The
last term in Eq.5 is a fixed systematic error associated with the DNN estimates which will be quantified
based on the MSE of the DNN prediction. Both $\frac{\partial PO_3}{\partial HCHO}$ and $\frac{\partial PO_3}{\partial NO_2}$ are derived from the sensitivity calculations
from Eq.3 divided by the satellite columns. All error terms in Eqs.6-9 are spatially and temporally invariant,
but the derivatives vary from pixel to pixel resulting in spatiotemporally-varying systematic errors.
Random errors originate from the uncertainty estimates coming with the TROPOMI and OMI L2
products and are somewhat reducible by averaging, and are given by:

$$e_{rand} = \sqrt{\left( \frac{\partial PO_3}{\partial HCHO} \cdot \gamma \cdot e_{rand-HCHO} \right)^2 + \left( \frac{\partial PO_3}{\partial NO_2} \cdot \gamma \cdot e_{rand-NO_2} \right)^2} \qquad (10)$$

where $e_{rand-HCHO}$ and $e_{rand-NO_2}$ are random retrieval errors. All terms in Eq.10 vary by time and location.
Table 1 summarizes the numbers used in the above equations and their origin.



**Table 1.** Values used in error calculations.

| Error terms | Systematic/Random | Value | Unit | Source |
|---|---|---|---|---|
| $e_{bc\text{-}NO2}$ and $e_{bc\text{-}HCHO}$ | Systematic | 0.01×VCD+0.06 | ×10$^{15}$ molec./cm$^2$ | Souri et al. (2025) |
| $e_{conv\text{-}HCHO}$ and $e_{conv\text{-}NO2}$ | Systematic | 0.09 | ppbv/(10$^{15}$ molec./cm$^2$) | Appendix B |
| $e_{DNN}$ | Systematic | 0.78 | ppbv/hr | Section 4.1 |
| $e_{rand\text{-}NO2}$ and $e_{rand\text{-}HCHO}$ | Random | Variable | ×10$^{15}$ molec./cm$^2$ | L2 Products |


It is important to acknowledge that the defined total error budget here is only a good guess and
optimistic. Some underlying sources of error, which are difficult to quantify, are not included. For example,
errors related to the training dataset derived from the F0AM model are challenging to assess because of the
lack of PO$_3$ measurements. We assume other inputs to the PO$_3$ parametrization, such as surface albedo and
H$_2$O(v), to be error-free. Additionally, all datasets used to estimate PO$_3$ contain spatial representation errors
(Souri et al. 2023), which are difficult to measure without knowing their true state of global spatial
variability. There are also assumptions regarding the equations mentioned earlier. For instance, it is assumed
that the validation of conversion factors can account for all systematic issues related to the vertical
distribution of NO$_2$ and HCHO in MINDS. Furthermore, we presume that the reported retrieval errors are
mostly random; however, this is not the case (Eskes et al., 2003; Boersma et al. 2018), and distinguishing
between these errors is not straightforward.
In case of oversampling of the PO$_3$ product both temporally and spatially, the total error will be given by:

$$e_{total} = \sqrt{\frac{1}{m}\sum e_{syst}^2 + \frac{1}{m^2}\sum e_{rand}^2} \qquad (11)$$

where *m* is the total number of samples. Eq.11 suggests that the systematic errors are persistent across all
samples and are not reducible by averaging, whereas the random errors become smaller by root square of
samples. In this equation, the assumption is that the root-mean-square of the systematic errors is a good
approximation of the systematic errors in the oversampled data because they are independent of each other.

## 4. Results and Discussion

In this section, we begin by validating and contrasting PO$_3$DNN against PO$_3$LASSO. Following
that, we use OMI to investigate the spatiotemporal variability of PO$_3$ and its sensitivity to photolysis rates,
HCHO, and NO$_2$ globally. We provide an application of the data to understand the effect of an extreme heat
wave on PO$_3$. Afterward, we offer a comprehensive global view of the PO$_3$ estimates algorithm by
integrating data from the TROPOMI compared with that one based on OMI. Finally, we document the total
error budget of the products.

### 4.1. *DNN performance*

We investigate the predictive power of the DNN algorithm against both validation and test data for
each air quality campaign or the entire aircraft dataset (Section 2.2). Figure 3 demonstrates the learning
curves (i.e., the evolution of MSE of prediction against the number of epochs corresponding to the number
of iterations of training the network for one cycle). All training datasets described in Section 3.1 are used
in this stage. Except for the early stages of training, both training and validation curves closely follow each
other, indicating that we possibly do not have overfitting issues. The curves are fairly smooth, resulting



from using the ADAM optimizer with a strictly small learning rate initially. Both curves converge to MSE
below 0.8 ppbv/hr which we use to assign the error of PO$_3$DNN prediction in Eq.5.

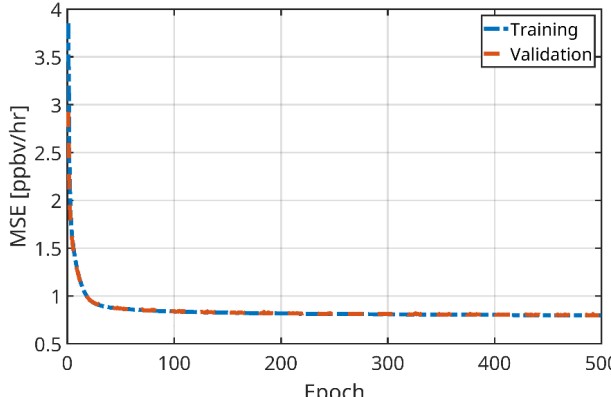


**Figure 3.** The learning curve shows the evolution of training and validation MSEs as a function of
epochs.

PO$_3$DNN has promising skill at predicting PO$_3$ across various atmospheric conditions. Figure 4
presents a comparison of the predicted PO$_3$ values against observationally-constrained F0AM values for
the test data for each suborbital mission. A similar comparison, which includes all data points measured
during each mission, can be found in Figure S1. The primary reason for highlighting the test data is that
they have never been used to fine-tune the DNN parameters. There is a strong correlation between the
predictions and the benchmarks across most campaigns for both the test data points (Figure 4) and the
complete set of aircraft measurements (Figure S1). Notably, the slope for the "All" test dataset is close to
the unity line. The DNN algorithm can reproduce over 96% of the variance in the test data. The model
performs significantly better than PO$_3$LASSO over INTEX-B compared to LASSO (as shown in Figure 7
in Souri et al., 2025). While the DNN's performance over the ATom campaigns is less impressive than in
other areas, it still represents a considerable improvement over LASSO, which was unable to reproduce
PO$_3$ in pristine regions ($R^2 < 0.05$). One key factor contributing to this improvement is the inclusion of
H$_2$O(v) in the input. Various parameters, including HO$_X$, are known to influence PO$_3$ in remote regions, but
these factors were not included in our parametrization. The method does not artificially inflate results by
introducing non-physical relationships in remote regions; the inability of the DNN to fully explain PO$_3$
during AToms suggests that it does not force unrealistic relationships between PO$_3$ and the inputs to
completely align with the F0AM results, leaving areas for future improvement in parametrization over
remote regions.




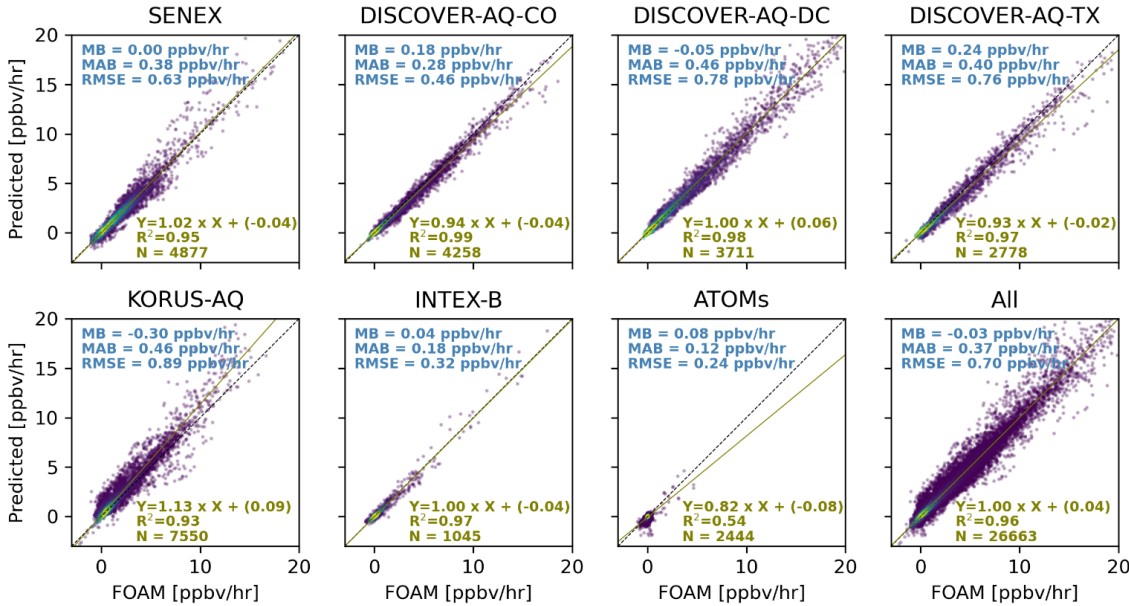

**Figure 4.** Scatterplots comparing observationally-constrained F0AM model PO₃ and the predictions that were based on the DNN for the test data from each air quality campaign. The test data have never been used for hyper tuning the algorithm. "All" denotes all test data.

*4.2.    Advantages of PO₃DNN over PO₃LASSO*

There are primarily four major benefits of PO₃DNN over PO₃LASSO that make the former parameterization a superior algorithm. The discussion of these advantages is as follows:

— *Higher predictive power*: PO₃LASSO predicted PO₃ for all datapoints collected from the suborbital missions with a $R^2=0.88$, RMSE=1.2 ppbv/hr, and a slope of 0.87 (Souri et al., 2025), whereas PO₃DNN reproduced the exact datapoints (Figure S1) with a $R^2=0.96$, RMSE=0.7 ppbv/hr, and a slope of 1.00. Furthermore, as shown in Figure 4, PO₃DNN has a great degree of generalization for datapoints outside of the training/validation data points. Consequently, these statistics suggest that DNN is a more powerful predictor.

— *Better representation of PO₃ over remote regions*: One notable limitation of PO₃LASSO was its inadequate representation of PO₃ in remote regions, such as during the ATOMS or INTEX-B campaigns. This led Souri et al. (2025) to entirely mask PO₃ estimates below 1 ppbv/hr. In these remote areas, PO₃ is typically influenced by the reactions between ozone and $HO_X$ in addition to $jO^1D$ and $H_2O$. While Souri et al. (2025) attempted to incorporate $H_2O$ into the LASSO parametrization, the algorithm assigned a zero coefficient to this parameter because of the use of the L1-regularization term. This term typically assigns a zero coefficient for a geophysical variable that is either irrelevant to the target or shows strong non-linear relationship with the target. PO₃LASSO did not factor in $H_2O(v)$ because $H_2O(v)$ exhibits a non-linear relationship with PO₃ – although the reaction between $O^1D$ and $H_2O$ can suppress ozone formation through the removal of $O^1D$, it produces two molecules of OH regenerating ozone in polluted places (Bates and Jacob, 2019). Consequently, the non-linear relationship between $H_2O$ and PO₃ is one that LASSO was unable to capture. While we could have addressed this by dividing the training dataset into different humidity levels (i.e., dry and humid), such an approach would have resulted in more discretization in the parametrization. Conversely, PO₃DNN can consider the non-linear





relationship between $H_2O$ and $PO_3$ without the need for empirical linearization. We observe a significant
improvement in predicted $PO_3$ for both AToms and INTEX-B campaigns compared to Souri et al.
(2025).

— *Diminished satellite error effects:* The reliance of $PO_3LASSO$ on FNR increases the contamination of
$PO_3$ predictions from satellite random noise. This primarily occurs because satellite errors associated
with HCHO and $NO_2$ adversely influence FNR (see Figure 12 in Souri et al. (2023a)), resulting in noise
in the empirical linearization approach used in $PO_3LASSO$. Even if we assume that all inputs to the
$PO_3LASSO$ parameterization, except for FNR, are error-free, the inherent randomness from choosing
among four different sets of equations segregated by the noisy FNR will still feed noise into the final
estimate. Although $PO_3DNN$ is inevitably influenced by satellite errors because of its dependence on
HCHO and $NO_2$ columns, it does not exacerbate these errors because it operates independently of FNR.
To demonstrate this tendency, Figure 5 shows the global $PO_3$ random error maps induced by OMI
HCHO and $NO_2$ retrieval random errors averaged in June 2006. We use identical inputs and errors for
both algorithms. Figure 5 is evidence of the diminished contamination of satellite random errors in
$PO_3DNN$ as compared to $PO_3LASSO$. The error differences tend to be larger over clean areas, because
FNR random errors are higher when both HCHO and $NO_2$ levels are small.

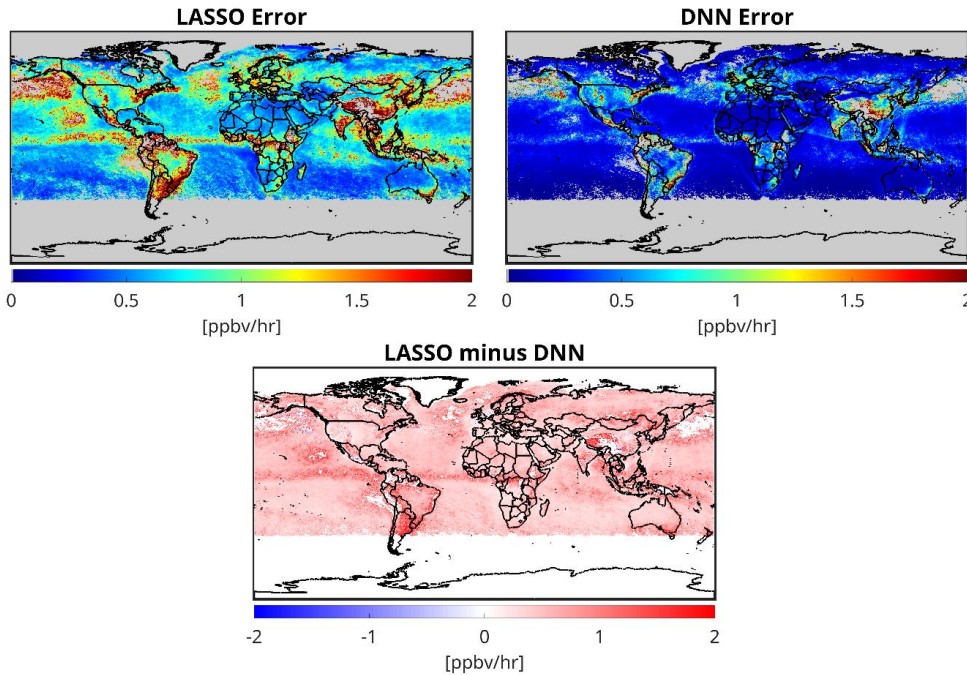

**Figure 5.** The comparison of the effect of satellite random errors in HCHO and $NO_2$ on $PO_3$ predictions
based on $PO_3LASSO$ and $PO_3DNN$ algorithms in June 2006. The data used for generating these maps are
based on OMI retrievals.

— *Continuity*: It is known that neural networks equipped with three hidden layers can well approximate
almost any high-dimensional non-linear function (Shen et al., 2021). An important superiority of
$PO_3DNN$ over $PO_3LASSO$ lies in the strength of the DNN algorithm at approximating high-
dimensional non-linear relationships between $PO_3$ and HCHO (a proxy for VOCR), $NO_2$ (a proxy for
reactive nitrogen), $jNO_2$ and $jO^1D$ (a proxy for photochemistry), and $H_2O$. While some of these non-
linearities were reasonably approximated in $PO_3LASSO$ by empirically segregating the chemical





conditions using FNR, the non-linear ozone photochemistry can go beyond the dependency on VOCs
and $NO_X$ levels. In fact, the relationship between $PO_3$ and VOCs and $NO_X$ can behave non-linearly
depending on the available light and water vapor as discussed in Section 4.3. This indicates that
traditional linear models, such as those using $VOCR/NO_X$ (or $HCHO/NO_2$) ratios, often fall short in
capturing this complexity because of the continuous and non-linear nature of these relationships.

### 4.3. *$PO_3$DNN can capture non-linear $PO_3$ chemistry as a function of pollution, light, and humidity*

To further elaborate on the capability of $PO_3$DNN to reasonably respond to variations in its five
major parameters in a mathematically continuous fashion, we create six isopleths, each specifically
designed to represent a particular atmospheric condition listed in Table 2. These isopleths are based on
perturbing HCHO and $NO_2$ in $PO_3$DNN and are shown in Figure 6.
It is immediately apparent that the hyperbolic shape of the $PO_3$ curve relative to $NO_2$ and HCHO
can be recreated by our algorithm, displaying a positive response to both HCHO and $NO_2$ on the right and
left sides of the ridgelines. This observation underscores the effective parametrization of the non-linearities
in ozone photochemistry achieved through the DNN algorithm. In the subplot representing normal
conditions, we overlaid three lines indicating FNR values of 1.5 (blue), 2.5 (green), and 3.5 (cyan). Souri
et al. (2025) used these lines to determine various coefficients in the $PO_3$LASSO parameterization. For
instance, the derivative of $PO_3$ with respect to $NO_2$ was determined to be -0.14 ppbv/hr for FNR < 1.5 but
increased to 6.54 ppbv/hr for FNR > 3.5. However, in practice, the thickness and curvature of the $PO_3$
isobars vary based on the prevailing atmospheric conditions, implying that the derivatives cannot
consistently retain the same values across the broad range of conditions.
In bright conditions, not only do we observe a significantly accelerated response of $PO_3$ compared
to the norm at identical $NO_2$ and HCHO concentrations, but the responses of $PO_3$ to these two compounds
also become more pronounced. Conversely, in dim conditions, both the magnitudes and responses are
weaker.
The contrast between dry and humid isopleths suggests that the presence of $H_2O(v)$ enhances $PO_3$
when abundant $NO_2$ and HCHO are present. This trend is similarly observed in the F0AM model, as
depicted in Figures S2, indicating that an increase in $H_2O(v)$ over polluted regions (arbitrarily defined as
$HCHO \times NO_2 > 10$) increases $PO_3$. Nonetheless, more humidity suppresses $PO_3$ especially where VOC is
limited and $NO_2$ is elevated possibly because the generated OH molecules from $O^1D + H_2O(v)$
predominantly react with elevated $NO_2$.
Lastly, we see the highest $PO_3$ rates recorded among all scenarios under a hypothetical condition
characterized by high humidity and photolysis rates. This condition is rare in nature because large amounts
of $H_2O(v)$ ($0.8 \times 10^{18}$) are confined to marine regions where surface reflectivity is low; nonetheless, an
intuitive tendency from $PO_3$DNN suggests that the algorithm does not create non-physical extrapolation
values.





**Table 2**. Six different atmospheric conditions defined to understand the response of $PO_3$ to HCHO
and $NO_2$ changes.

| Labels | $H_2O$ [molec/m$^3$] | $jO^1D$ [1/s] | $jNO_2$ [1/s] | Notes |
|---|---|---|---|---|
| *Norm* | $0.4\times10^{18}$ | $4\times10^{-5}$ | $1.2\times10^{-2}$ | A typical condition in summer in the eastern US at noon |
| *Bright* | $0.4\times10^{18}$ | $7\times10^{-5}$ | $1.4\times10^{-2}$ | Central America with abundant sunshine in the afternoon |
| *Dim* | $0.4\times10^{18}$ | $3\times10^{-5}$ | $0.7\times10^{-2}$ | Scandinavia in the afternoon summer |
| *Dry* | $0.1\times10^{18}$ | $4\times10^{-5}$ | $1.2\times10^{-2}$ | An arid region such as Spain Meseta Central in the afternoon summer |
| *Humid* | $0.8\times10^{18}$ | $4\times10^{-5}$ | $1.2\times10^{-2}$ | A place the like Persian Gulf with high humidity and abundant sunshine |
| *Humid and Bright* | $0.8\times10^{18}$ | $7\times10^{-5}$ | $1.4\times10^{-2}$ | Since accelerated photolysis rates close-to-surface usually occur over bright regions (arid) with low humidity, this condition is rare in nature. |


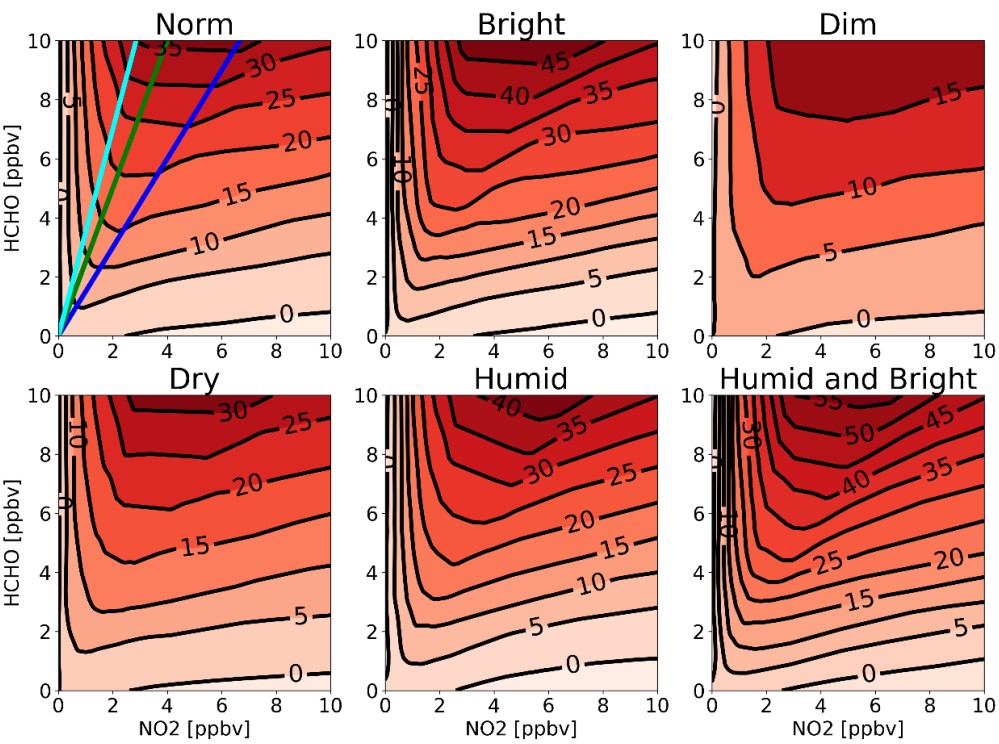




**Figure 6.** The contour maps of PO$_3$ isopleth generated by PO$_3$DNN algorithm for six different atmospheric conditions defined in Table 1. In the first subplot, blue, green, and cyan lines indicate FNR=1.5, 2.5, and 3.5, respectively. Numbers on isobars are in ppbv/hr.

### *4.4. PO$_3$ Maps using OMI and TROPOMI: A General View and Anomalous Changes*

*4.4.1. Global PO$_3$ and Seasonality using OMI in 2005-2007*

Figure 7 shows the global distribution of PO$_3$ rates averaged over a quarter-degree in 2005-2007, using OMI HCHO and NO$_2$ retrievals. It also includes whisker-box plots highlighting seasonal variations in PO$_3$ for selected regions and cities. We selected the 2005-2007 timeframe for this analysis because the OMI data were free from degradation issues, including the row anomaly. The map indicates accelerated PO$_3$ rates across heavily polluted regions, such as cities in the Middle East, Asia, the U.S., Central Europe, and Africa, aligning with what we observed in Souri et al. (2025). While some areas exhibit significant seasonal fluctuations, others show little variability throughout the seasons. Notably, the east coast of the U.S., Central Europe, China, Tehran, and Johannesburg experience peak PO$_3$ rates in summer. This pattern is primarily attributed to enhanced photochemistry and the elevated sensitivity of PO$_3$ to NO$_X$, driven by increases in VOCR/NO$_X$ (Souri et al., 2025).

The seasonal variability of PO$_3$ in two African regions, characterized by biomass burning, exhibits an anti-correlation. This occurs because biomass burning in the northern hemisphere of Africa occurs from November to March, while the southern hemisphere in Africa experiences it from June to September (Roberts et al., 2009). Southeast Asia also shows a peak in PO$_3$ during the biomass burning season (August-September).

Places like Mexico City, several major Brazilian cities (including Sao Paulo and Rio de Janeiro), northern India, and the southwest coast of the U.S. show minimal seasonal variability in PO$_3$. The lack of pronounced seasonal changes may be attributed to less pronounced fluctuations in photolysis rates or substantial spatial heterogeneity in the seasonal variabilities of HCHO and NO$_2$, resulting in reduced seasonal variations but with greater variance. Nonetheless, certain weather conditions can influence these results; for instance, monsoon flows can disperse and scavenge pollution from the northern India around July-September (David and Nair, 2013), dampening PO$_3$. Mexico City also experiences a monsoon season in summer causing pollution to subside temporarily. The attribution of the seasonality will be discussed in the next section.



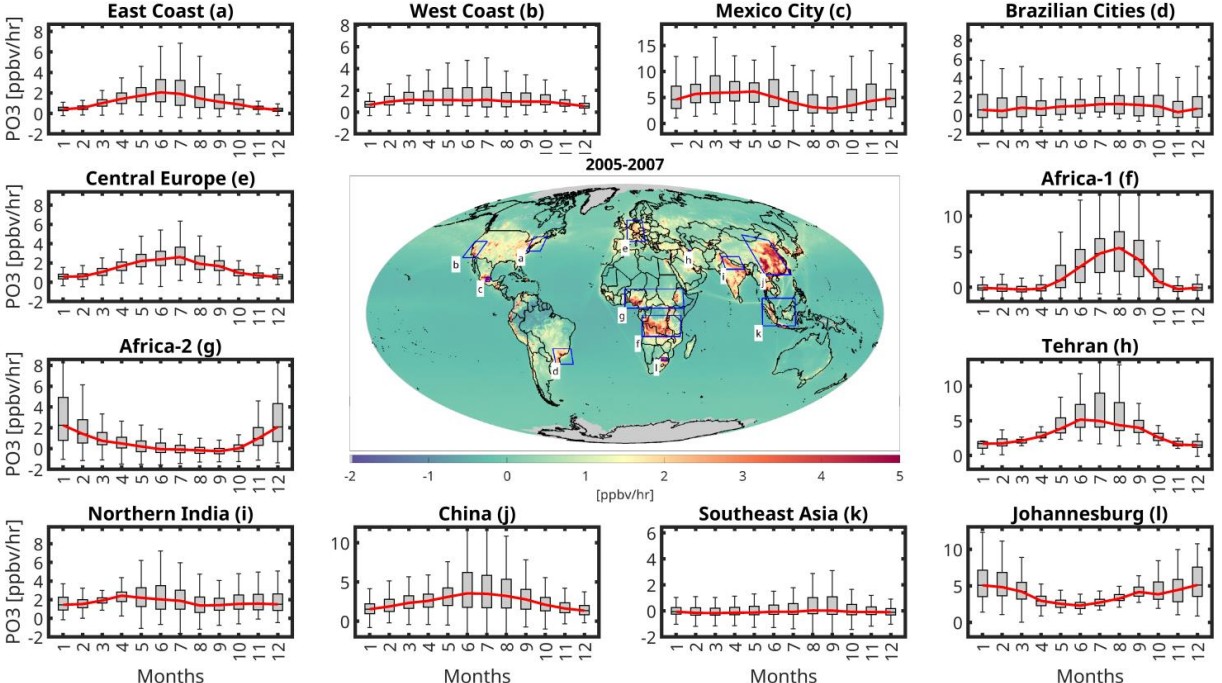

**Figure 7.** (center) The averaged global PO₃ map at 0.25°×0.25° in 2005-2007 based on the new algorithm. OMI data are used to populate HCHO and NO₂ abundance. (margins) the whisker-box plots of PO₃ seasonality over various selected regions. In the box plot, the central red line shows the median, and the top and bottom edges of the box show the 25th (q1) and 75th (q3) percentiles. The dark solid lines at the very beginning and the end of each plot show the minimum and maximum values excluding the outliers. The outliers are removed based on by any value above q3+1.5×(q3−q1) or below q1−1.5 × (q3−q1).

*4.4.2. The attribution of PO₃ seasonality*

Photolysis rates, which serve as crucial indicators of photochemical activity, are the primary determinants of PO₃ seasonality. Figure 8 illustrates the sensitivity of PO₃ to NO₂, HCHO, and combined J-values (jNO₂ and jO¹D) based on Eq.3 across the same regions and months presented in Figure 7. The absolute values of PBL HCHO, NO₂, and jNO₂ are shown in Figure S3. As shown in Appendix A, these sensitivity values are influenced by both the magnitude of the precursor and the first derivative of PO₃ with respect to that precursor. Thus, the sensitivity values should be interpreted as the result of these combined effects.

The amplitude of photolysis rates dictates the amplitude of the sensitivity of PO₃ to NO₂ and HCHO. For instance, over East Coast, Central Europe, and Tehran, the first derivative of PO₃ to NO₂ tends to be small during colder months, primarily because of reduced photochemistry and non-linear chemistry. As a result, despite significantly higher NO₂ concentrations in these months, the sensitivity of PO₃ to NO₂ is muted; this tendency indicates that the derivative effect can overshadow the increase in NO₂ concentrations. Conversely, in warmer months, the larger positive derivative of PO₃ relative to NO₂, driven by increased HCHO levels (shifting away from VOC-sensitive regimes) and enhanced photolysis rates, markedly increases the contributions of low summer NO₂ levels to PO₃. Likewise, we observe substantially higher sensitivity of PO₃ to HCHO concentrations during warmer seasons. This increase is attributed to




both the elevated levels of HCHO and the growing derivative of $PO_3$ with respect to HCHO, both of which
are directly influenced by enhanced photochemistry. One might argue that summer conditions should lead
to a shift towards extremely $NO_X$-sensitive regimes, resulting in a reduced first-order derivative of $PO_3$ to
HCHO. However, most polluted regions chosen for this figure are in transitional regimes during the
summer, which renders $PO_3$ fairly responsive to HCHO concentrations. These results underscore the
importance of including photolysis rates in ozone sensitivity analysis, rather than relying solely on FNR in
former studies. For example, a lower FNR in the morning (~0930 LST) compared to the afternoon may
wrongly suggest that $PO_3$ would become more sensitive to VOCs earlier in the day. However, decreased
light in the morning reduces the sensitivity of $PO_3$ to VOCs, despite a lower FNR.

The sensitivity of $PO_3$ to photolysis rates is dependent on pollution levels, just as its sensitivity to
HCHO and $NO_2$ concentrations is influenced by photolysis rates. This is primary reason for seeing minimal
seasonality of $PO_3$ over Mexico City, various Brazilian cities, and northern India. These minimal changes
in photolysis rate sensitivities are caused by the less pronounced seasonality in both photolysis rates and
pollution levels compared to other areas (Figure S3). Souri et al. (2025) found that photolysis rates
significantly contribute to the production of $PO_3$ when there is an adequate amount of ozone precursors.
This was reflected in larger coefficients associated with photolysis rates in $PO_3$LASSO algorithm for
FNR<1.5, where pollution levels were high. For example, high photolysis rates over the Sahara do not
significantly contribute to $PO_3$ because of the limited availability of ozone precursors needed to initiate the
$RO_X$-$HO_X$ cycle. A notable example can be observed in Africa, where photolysis rates tend to remain
consistent throughout the year under near cloud-free conditions (Figure S3). However, there is a marked
seasonality in the sensitivity of $PO_3$ with respect to photolysis rates during polluted months suggesting that
the ample precursors can leverage available lights to form more ozone molecules. This pattern underscores
the algorithm's capability to understand the intertwined relationships between the photolysis rate
sensitivities and pollution levels, as well as the pollution sensitivities and photolysis rates.

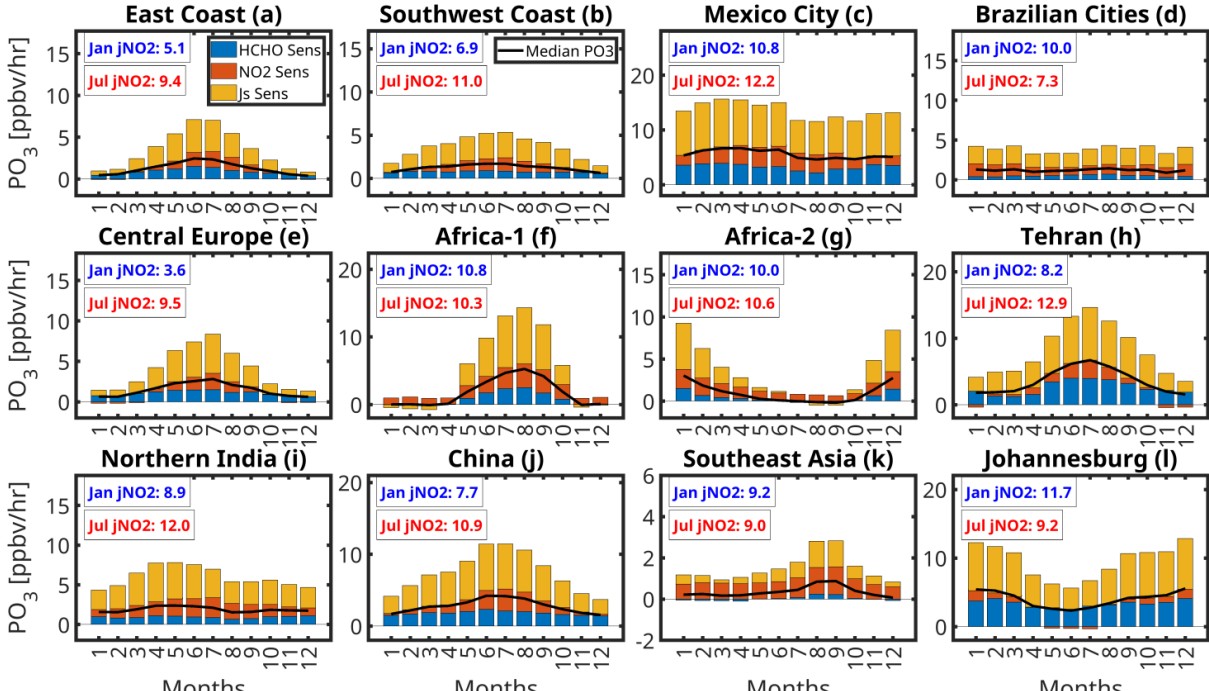



**Figure 8.** The bar plots of the sensitivity of $PO_3$ to photolysis rates, $NO_2$, and HCHO concentrations within the PBL over the selected regions shown in Figure 7. These sensitivities are influenced by both the magnitude of the precursors and the first-order derivative of $PO_3$ to the precursor, detailed in Appendix A. $jNO_2$ values are in $1 \times 10^3$/s units.

### 4.4.3. Can rapid heatwaves accelerate $PO_3$ in the northeast U.S.?

Heatwaves are known to worsen ozone pollution in many regions because of several factors. Increased temperatures can elevate both anthropogenic and biogenic VOCs (Guenther et al., 2012; Wu et al., 2024). Additionally, higher temperatures can accelerate some key reaction rates, particularly $NO+RO_2$ (Pusede et al., 2015). Longer periods of active photochemistry can occur because of fewer clouds, and the dispersion of ozone and its precursors may be hindered by a dominant high-pressure system (Pyrgou et al., 2018). However, some of these effects may be offset if heatwaves last for an extended period, as drought conditions can suppress biogenic VOCs (Duncan et al., 2009; Demetillo et al., 2019). In this study, we focus on a severe heatwave that affected the eastern U.S. in August 2007. To contrast the atmospheric conditions during this month with those of a typical condition, we use August 2008 as a baseline.

Our $PO_3$DNN product shown in Figure 9 exhibits a 21% increase in $PO_3$ rates with respect to the baseline throughout the northeast U.S., suggesting that rapid heatwaves can accelerate the production of chemically-generated ozone molecules. It is important to acknowledge that both maps represent conditions with minimal cloud cover imposed by the cloud-screening flags from the satellite retrievals. However, the frequency of clear-sky conditions is usually higher during heatwaves compared to normal conditions. This distinction is critical because clouds play a significant role in reducing photochemical activity close to the surface by limiting incoming solar radiation. Consequently, even if $PO_3$ values appeared similar between these two episodes, the more frequent occurrence of clear-sky conditions in August 2007 would result in a greater cumulative contribution of $PO_3$ to the region. This highlights the impact of persistent sunshine in enhancing ozone, reinforcing the need to account for meteorological variability when analyzing photochemical processes.

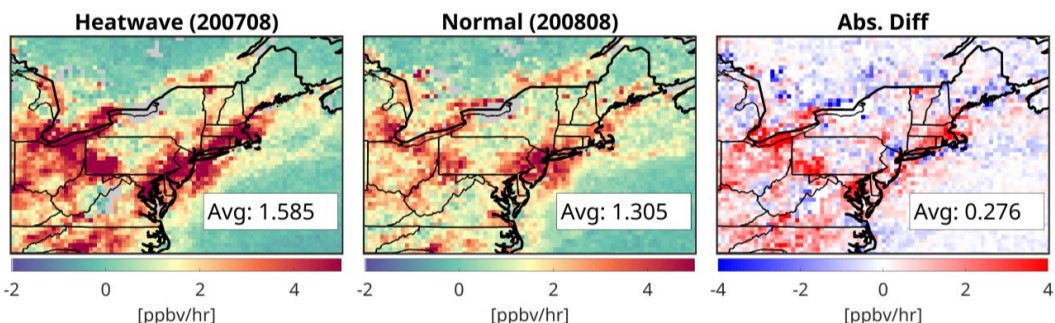

**Figure 9**. The maps of $PO_3$ within PBL in August 2007 (left), August 2008 (middle), and their absolute difference (right).

To study the reasons behind the accelerated $PO_3$ during this episode, we explore the respective changes in ozone precursors and the sensitivities by the heatwave. Figure 10 contrasts the differences in $NO_2$ and HCHO levels within the PBL region for two episodes. These maps are derived from the bias-corrected OMI VCDs scaled to the PBL region using MINDS simulations. Different wind patterns are most likely the cause of the differences in $NO_2$ patterns over cities; we see different shapes of $NO_2$ plumes over NYC, Toronto, and Boston. Additionally, we see some uniform enhancements of $NO_2$ in several inland regions, such as Washington DC, Philadelphia, North Carolina, Tennessee, and Ohio. While we cannot definitively locate the cause of these enhancements without additional measurements and models, we can speculate that rising temperature can increase both nitrification and denitrification microbial activities under



optimal soil moisture causing soil $NO_X$ emissions to rise. Another possible explanation could be that $NO_X$
reservoirs (such as PAN and alkyl nitrate) can rapidly be converted back to $NO_2$ because of higher
temperature and more sunshine. HCHO levels are markedly high during the heatwave event in comparison
to the baseline (>2 ppbv). The enhanced biogenic emissions and photochemistry are the causes.

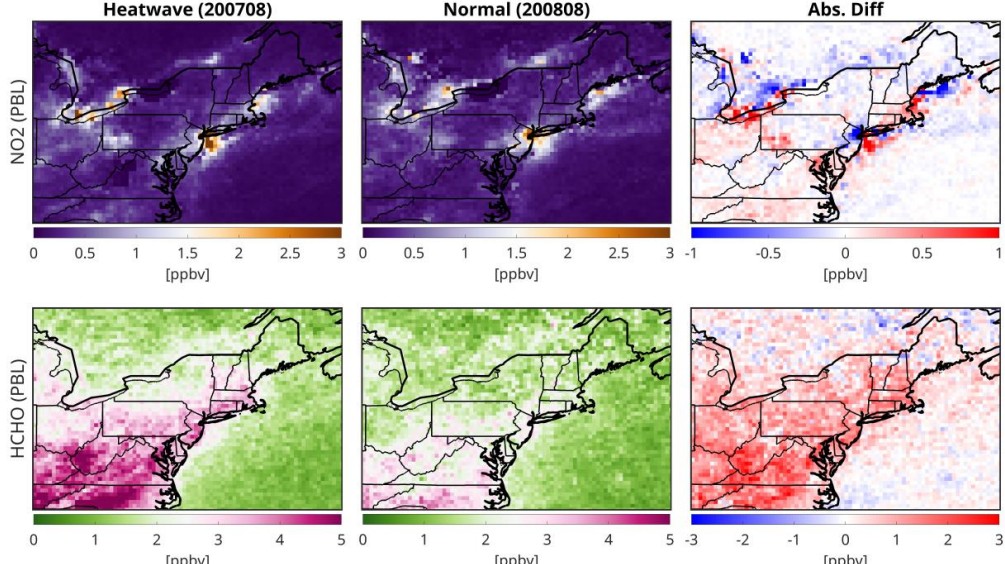

**Figure 10.** The maps of $NO_2$ (top) and HCHO (bottom) concentrations within PBL averaged in month of
August in 2007 (heatwave) and 2008 (a normal condition), and their absolute differences. The abundance
of HCHO and $NO_2$ are informed by bias-corrected OMI retrievals.
Using the spatially-varying sensitivity of $PO_3$ to $NO_2$ and HCHO provided by our product, we find
that in rural and suburban areas, the derivative of $PO_3$ to $NO_X$ tends to be high ($NO_X$-sensitive regimes)
(Figure 11), indicating that even a small increase in $NO_2$ in several inland regions can boost the sensitivity
of $PO_3$ to $NO_2$ greatly. This observation is consistent with findings from Geddes et al. (2022), who reported
a similar trend of increasing ozone sensitivity to soil $NO_X$ emissions across various remote regions in the
U.S.
We observe that higher levels of HCHO significantly increase the sensitivity of $PO_3$ to HCHO in
several high-$NO_X$ areas, including Toronto, Boston, Washington DC, Philadelphia, and Buffalo, as
illustrated in Figure 11. As noted before, the sensitivity maps are influenced by both the magnitude of a
precursor and the derivative of $PO_3$ with respect to a precursor. In high-$NO_X$ regions, the derivative of $PO_3$
to HCHO is typically large (i.e., VOC-sensitive). Consequently, elevated HCHO concentrations lead to a
greater sensitivity of $PO_3$ to HCHO levels. Conversely, in remote regions where the derivative of $PO_3$ to
HCHO is small, increases in HCHO cannot induce noticeable effect on $PO_3$. An exception to the general
increase in $PO_3$ sensitivity to HCHO occurs over NYC. This anomaly can be attributed to the different
shape of the $NO_2$ plume in August 2008 as compared to 2007. In August 2007, as shown in Figure 10, $NO_2$
concentrations in NYC were dispersed over the ocean, resulting in less VOC-sensitive conditions (lower
derivative of $PO_3$ to HCHO) within the city. As a result, $PO_3$ sensitivity values to HCHO decrease because
the first-order derivative decreases.



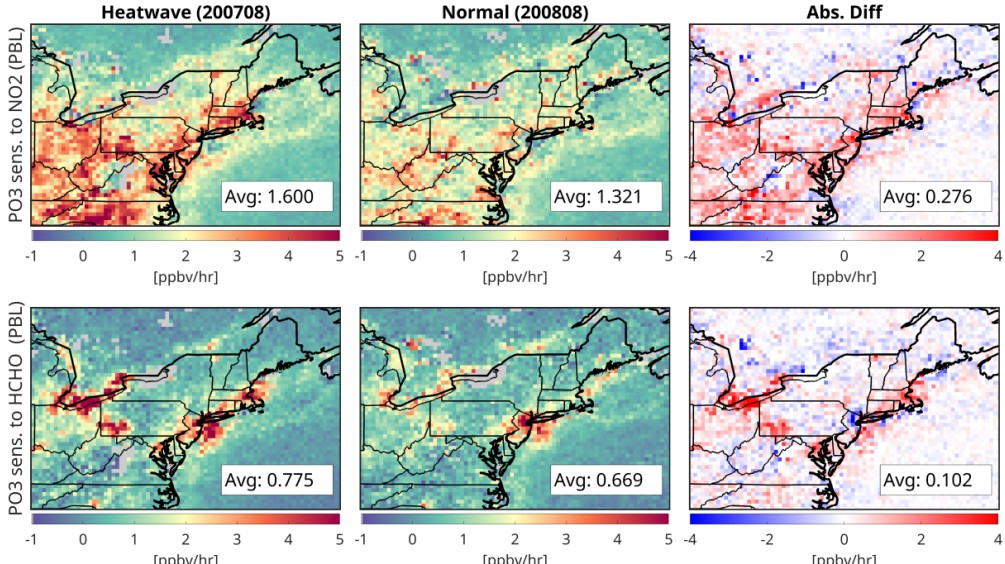

**Figure 11.** A similar layout as shown in Figure 10, but with the sensitivity outputs derived from the
PO$_3$DNN algorithm.
*4.4.4. High resolution TROPOMI-based PO$_3$ maps and sensitivities contrasted with OMI*

Accelerated rates of PO$_3$ at approximately 1330 LST are observed consistently across polluted
midlatitude regions characterized by high photolysis rates. This pattern is substantiated by the global PO$_3$
maps derived from TROPOMI and OMI data for the year 2019 illustrated in Figure 12. While the maps
presented are averages for 2019, significant PO$_3$ hotspots (exceeding 8 ppbv/hr) are identified over
metropolitan/industrial areas including Mexico City (Mexico), Tehran (Iran), the Persian Gulf, and Hunan
Province (China). There are less documented regions undergoing elevated locally-produced ozone such as
Johannesburg (South Africa), Rio de Janeiro (Brazil), Sao Paulo (Brazil), and Santiago (Chile). In contrast,
Europe emerges as a region with comparatively low PO$_3$ levels despite its dense population. This tendency
may be attributed to lower photolysis rates (characterized by high solar zenith angles and low surface
reflectivity) as well as effective emissions mitigation strategies. A notable similarity exists between these
identified hotspots and those reported by Souri et al. (2025), although the contrast between clean and
polluted areas is more pronounced in the PO$_3$DNN product because of an improved representation of
PO$_3$DNN in clean regions.

PO$_3$ exhibits a slight negative value over oceanic and densely forested areas (such as the Amazon
and Congo), primarily because of ozone sinks associated with water vapor (H$_2$O(v)) and alkenes, which are
implicitly included in our parametrization. However, a marked contrast is observed between the slightly
negative and positive PO$_3$ levels along marine vessel pathways. These ship paths are informed not only by
remote sensing data (Georgoulias et al., 2020) but also by the conversion of column measurements to PBL
mixing ratios thorough the MINDS simulation, which accounts for ship emissions. Given that the PBL is
typically shallow over marine regions, the conversion factor is expected to be substantial for these
pathways, resulting in a pronounced contrast in pollution levels within the PBL.

The finer spatial resolution of the TROPOMI dataset enhances the detail of the PO$_3$ maps compared
to those derived from OMI, yielding less noise and fuller data. This reduction in gaps in TROPOMI-based
PO$_3$ is attributed to a lower likelihood of cloud contamination and the full coverage of all detectors, in





contrast to OMI, which suffers from the row anomaly. Visual analysis of the two datasets indicates that
TROPOMI consistently shows higher $PO_3$ than OMI over polluted regions. Except for $NO_2$ and HCHO
VCDs, the inputs to the parametrization are identical across both products.

To further investigate these differences, we synchronized the TROPOMI datasets at the OMI-based
spatial resolution and produced scatterplots, as displayed in Figure 13. The correspondence between the
two products is high ($R^2 = 0.86$). Nonetheless, TROPOMI-based $PO_3$ levels are approximately 10% greater
than those derived from OMI. The fact that we observe this overestimation given that TROPOMI has been
coarsened to match OMI's footprint suggests that the differing spatial resolutions (0.25 degrees versus 0.1
degrees) are unlikely to account for the discrepancy. Moreover, we undertake a comparative analysis of
$NO_2$ and HCHO mixing ratios within the PBL region as obtained from MINDS alongside these two satellite
datasets. Given that the conversion factor remains consistent between the two products, any observed
differences can be attributed to variations in their respective VCDs. Our analysis reveals that both $NO_2$ and
HCHO mixing ratios are higher in TROPOMI relative to OMI (by 5-6%), thereby providing a solid
explanation for the elevated TROPOMI-based $PO_3$ in comparison to OMI.

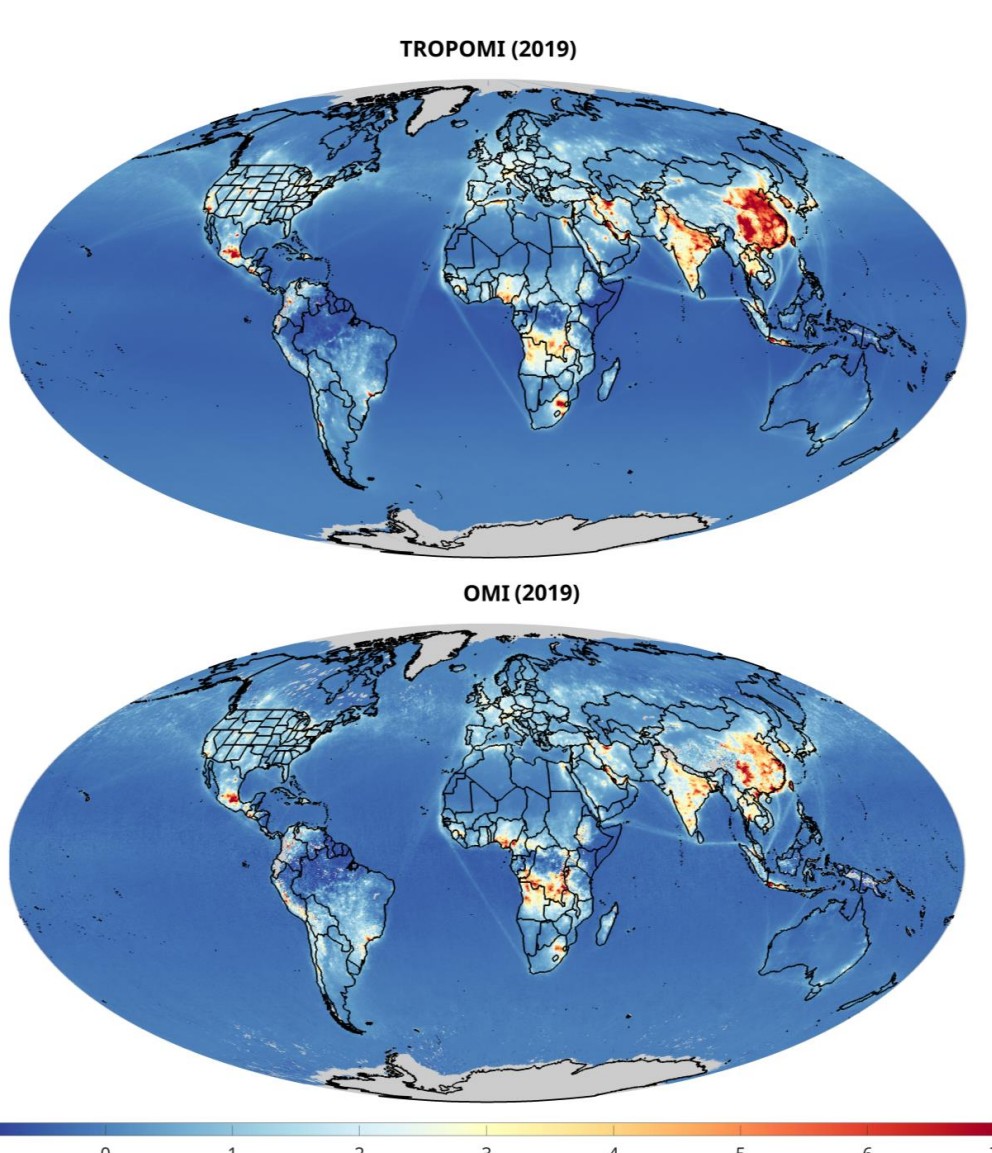

**Figure 12**. Global maps of PO$_3$ derived from TROPOMI (top) and OMI (bottom) datasets based on the PO$_3$DNN algorithm in 2019. These values are estimated within the PBL region at ~1330 LST. The data exclude cloudy pixels, strong smoke, sensor anomalies, and snow based on the recommended quality flags coming with TROPOMI and OMI products.



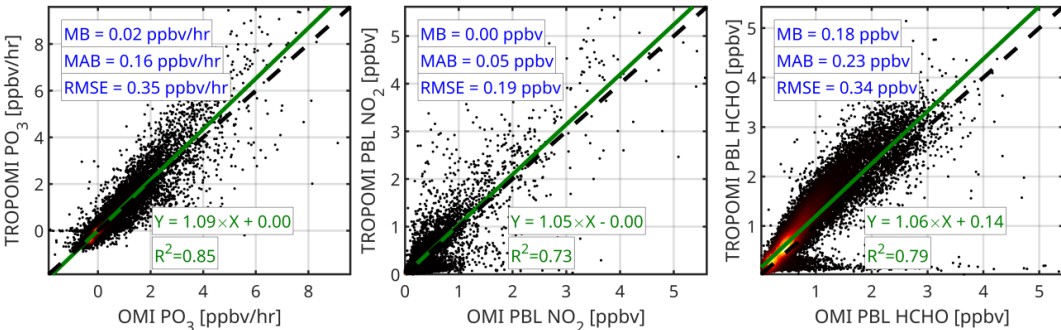

**Figure 13**. Scatterplots of (left) OMI $PO_3$ vs. TROPOMI $PO_3$, (middle) OMI PBL $NO_2$ vs. TROPOMI PBL
$NO_2$, and (right) OMI PBL HCHO vs. TROPOMI PBL HCHO based on 2019. We coarsen TROPOMI
dataset to match OMI's spatial resolution to remove the effect of spatial footprint on these results.

We explore the spatially varying sensitivity of $PO_3$ to HCHO and $NO_2$ worldwide. These maps
provide finer information compared to binary maps obtained from FNRs. Figure 14 illustrates global maps
of these sensitivities averaged for the year 2019. We observe negative sensitivity values of $PO_3$ to $NO_2$ in
urban areas, which aligns with our understanding of non-linear ozone chemistry. These negative values are
particularly pronounced in northern China, where VOC/$NO_X$ ratios remain low throughout the year. Similar
non-linear feedback patterns can be seen in the Benelux region and the United Kingdom, primarily driven
by elevated $NO_2$ levels. In contrast, $NO_2$ significantly contributes to higher $PO_3$ levels in southern China,
India, Mexico, and several regions across Africa.

As indicated in Souri et al. (2025), the influence of HCHO on $PO_3$ is largely governed by $NO_X$
emissions. This relationship explains why the sensitivity of $PO_3$ to HCHO closely mirrors global $NO_2$ levels,
which dictates the locations of VOC-sensitive regimes. We observe slightly negative sensitivity of $PO_3$ to
HCHO in remote and densely vegetated regions, likely a result of the effects of alkenes on ozone. However,
the implicit nature of DNN makes it challenging to identify the exact chemical reasons behind these
patterns. Noteworthy examples of areas where $PO_3$ is significantly influenced by HCHO include eastern
China, Los Angeles (USA), Tehran (Iran), Mexico City (Mexico), and Johannesburg (South Africa).



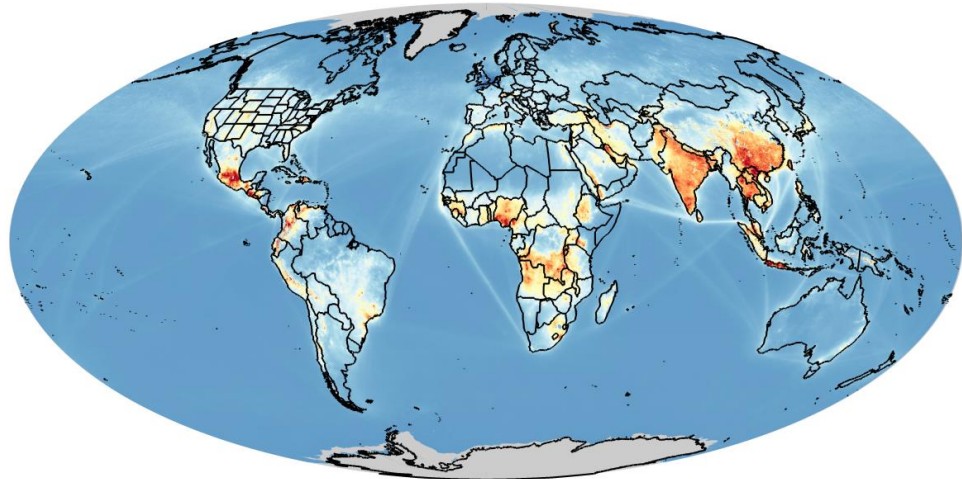

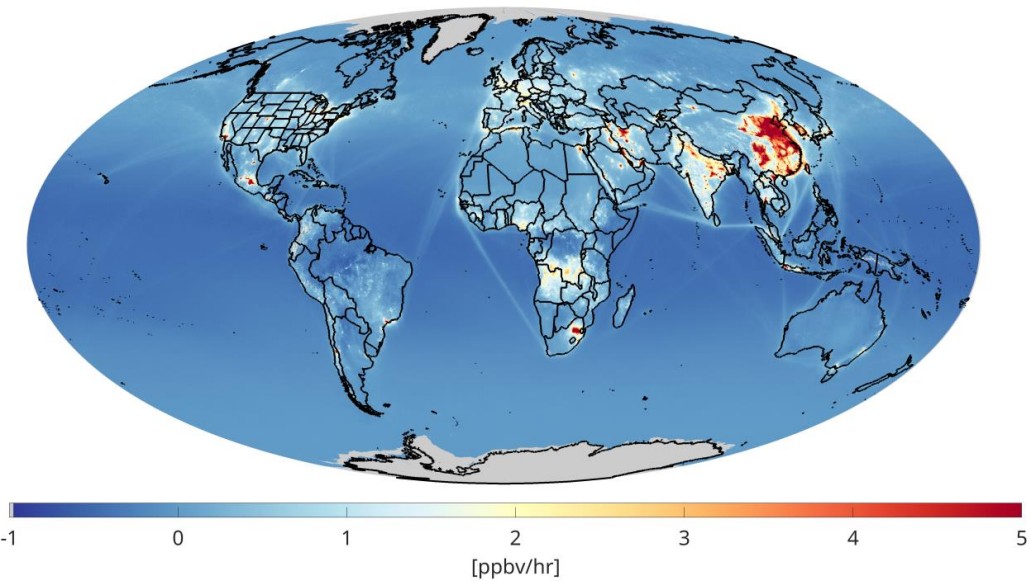

**Figure 14.** The sensitivity of PO$_3$ to NO$_2$ (top) and HCHO (bottom) based on our algorithm using TROPOMI data in 2019.




Figure 15 presents the maps of PO$_3$ along with sensitivities across four seasons in 2019 over Middle
East, derived from TROPOMI data. Notably, PO$_3$ values surge during the summer months in several densely
populated and industrial regions of the Middle East. Furthermore, we observe considerable PO$_3$ values in
the fall, primarily caused by the influence of HCHO. This fall peak is consistent with the observations made
by Souri et al. (2025), who reported a sharp rise in PO$_3$ in late fall 2019 over Tehran (Iran). The overall
seasonality of PO$_3$ is well aligned with the discussions presented in Section 4.4.1. The sensitivity of PO$_3$ to
NO$_2$ exhibits notable variation, shifting from low and negative values during the colder months to positive
and high values in the warmer months. We identify HCHO as the predominant contributor to PO$_3$ in these
regions, as the majority of these cities fall in VOC-sensitive environments and emit significant amounts of
anthropogenic HCHO, whether from primary or secondary sources. These maps eliminate the need for
binarization of chemical conditions, as they effectively illustrate the spatial variability in ozone response to
HCHO and NO$_2$.

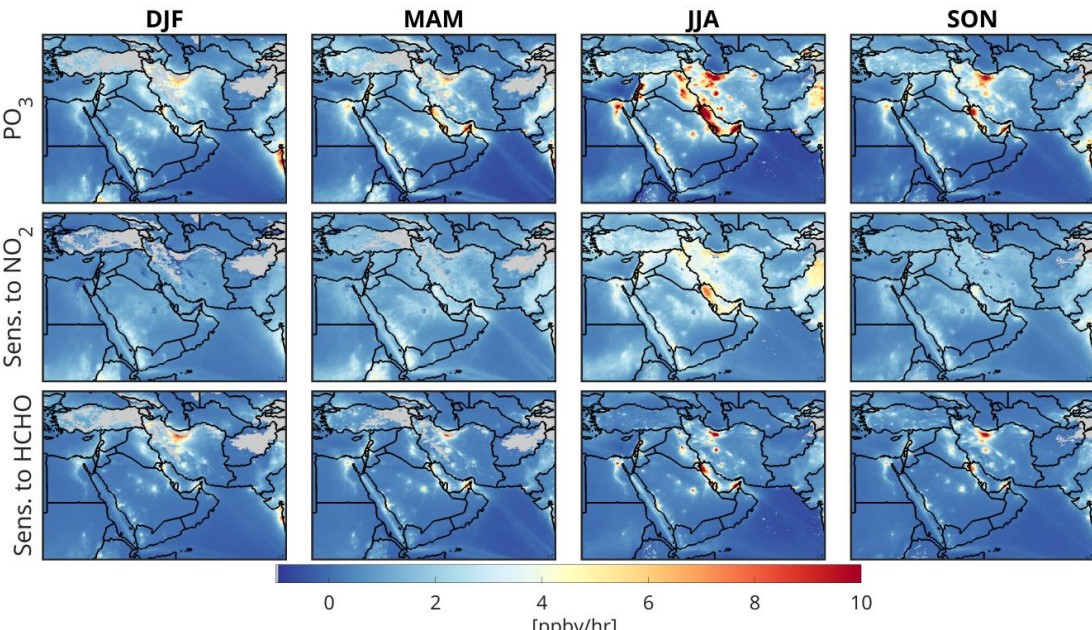

77

**Figure 15.** The magnitude of PO$_3$ and the corresponding sensitivity to NO$_2$ and HCHO over Middle East
grouped into four different seasons. DJF: December-January-February, MAM: March-April-May, JJA:
June-July-August, and SON: September-October-November. *Sens.* means sensitivity.

*4.4.5. A Tale of Two Cities: Long-term trends of PO$_3$ in Los Angeles vs. Tehran in 2005-2019*
Using a linear trend calculation method outlined by Souri et al. (2024), we evaluate the long-term
linear trends of PO$_3$ maps in two cities: Los Angeles (USA) and Tehran (Iran). Figure 16 clearly
demonstrates a complete reversal in the linear trends of PO$_3$, revealing an increase in Los Angeles and a
decrease in Tehran. Moreover, we observe similar contrasting trends in the surrounding areas, with PO$_3$
levels rising near Tehran while declining near LA (Los Angeles). This tendency is a textbook example of
non-linear ozone chemistry. While we do not identify any statistically significant trends in HCHO mixing
ratios within the PBL for these two major cities, we do observe a significant downward trend in NO$_2$ mixing
ratios in Los Angeles and a substantial upward trend in Tehran, as illustrated in Figure 17. Since both cities
are primarily in VOC-sensitive conditions at their cores (Souri et al., 2025), it is intuitively clear a reduction
(enhancement) in NO$_2$ should lead to positive (negative) trends in PO$_3$ because of the impact of the loss of





NO$_X$ on PO$_3$. Conversely, in their suburbs where the negative effect of the loss of NO$_X$ on PO$_3$ diminishes,
we see a close association of the sign of PO$_3$ trends and those of NO$_2$.

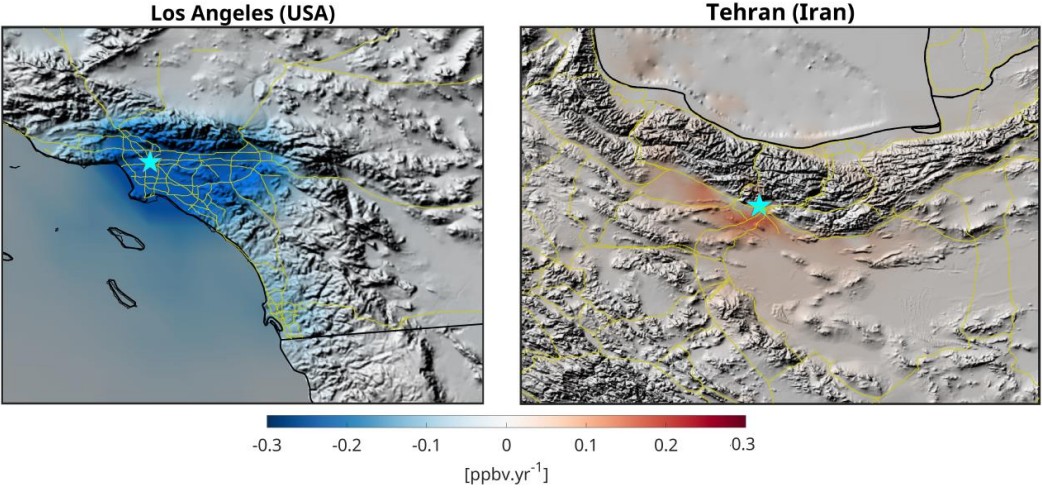

**Figure 16**. The statistically significant linear trends of PO$_3$ over LA (left) and Tehran (right) based on the
PO$_3$DNN product in 2005-2019.

**Figure 17**. The statistically significant linear trends of PBL NO$_2$ mixing ratios over LA (left) and Tehran
(right) based on the bias-corrected OMI and MINDS product in 2005-2019.

### 4.5. *Error Analysis*

Based on the formulation outlined in Section 3.4, we evaluate both the systematic and random error
components of PO$_3$ for July 2019, based on data from both OMI and TROPOMI retrievals. Figure 18
presents the average error values for the month. Total PO$_3$ errors range from 25% to 80% in areas
characterized by moderate to extreme pollution, while in more remote regions, errors can surpass 200%.



On average, random errors constitute only a small fraction of the total error budget, with OMI showing consistently larger random errors than TROPOMI across the region. This is primarily a result of OMI's limited sampling caused by row anomaly issues. As mentioned in Section 4.2, these random errors are significantly lower when compared to the PO$_3$LASSO random errors (Souri et al., 2025).

Systematic errors account for most of the total error, exceeding 90%. These systematic errors are comprised of three components: biases arising from the correction of VCDs using ground-based remote sensing data, errors related to DNN predictions, and conversion factors derived from the MINDS framework. The first two components contribute minimally to the overall error (less than 5%), making the MINDS conversion factors the dominant contributor to the total error budget. Therefore, any parametrization aimed at converting satellite-based VCDs to near-surface concentrations, including the one presented in this study, should always seek out a model that accurately reflects the shape of the profiles.

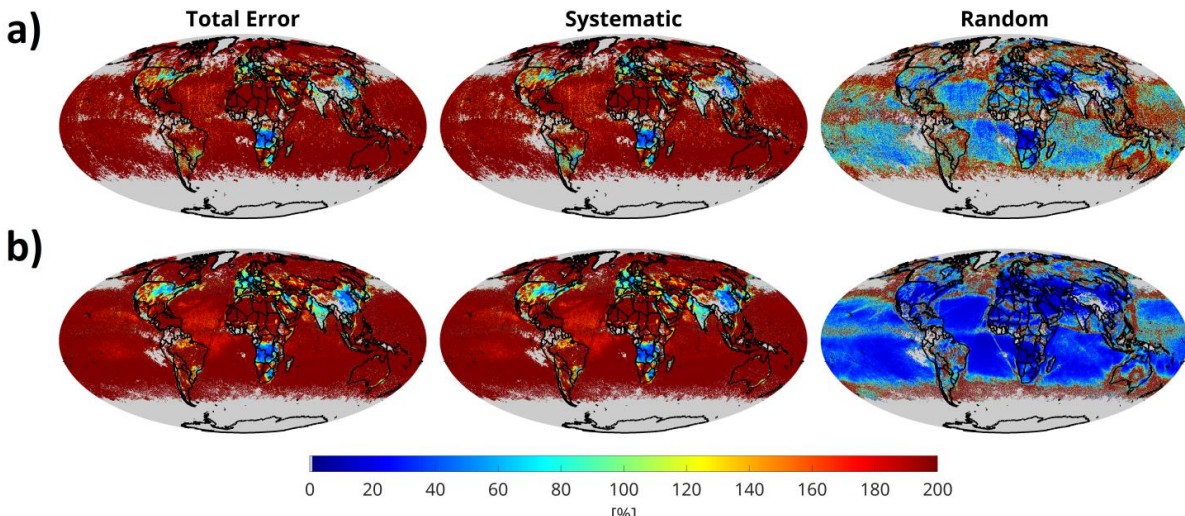

**Figure 18.** The maps of total error, systematic, and random errors for (a) OMI, and (b) TROPOMI computed for July 2019.

## 5. Summary

Early data-driven analyses of ozone chemistry sensitivity primarily relied on "ratio-based" indicators to partially linearize the non-linear aspects of urban ozone chemistry, which are influenced by pollution levels, light, and water vapor. With the development of more sophisticated algorithms, including machine learning techniques capable of fitting high-dimensional non-linear functions, we have shown that a highly effective parameterization of net ozone production rates (PO$_3$) can be achieved. This approach not only eliminates the need for empirical linearization of ozone chemistry through various indicators, but it also allows for the primary inputs to be accurately constrained using satellite observations. This advancement allowed us to move beyond the previously employed formaldehyde-to-nitrogen dioxide ratio (FNR) and to generate more comprehensive sensitivity maps, which account for variations not only in HCHO and NO$_2$ but also in light and water vapor.

We significantly enhanced the empirical parametrization of PO$_3$ described in Souri et al. (2025), in several key ways: (i) we improved the representation of PO$_3$ in both polluted and clean areas using a L2-regularized deep neural network (DNN) and eliminated the need for empirical linearization of atmospheric conditions with the FNR approach, resulting in reduced complexity and noise in the final estimates; (ii) we



used a finer, up-to-date global transport model called MINDS to convert satellite-retrieved vertical column density (VCD) into planetary boundary layer (PBL) mixing ratios; (iii) we incorporated the error from these conversion factors, derived from comprehensive validation against aircraft spirals, into the total error budget; and (iv) we generated long-term records of $PO_3$ magnitudes and sensitivities to nitrogen dioxide ($NO_2$) and formaldehyde (HCHO) using bias-corrected data from the Ozone Monitoring Instrument (OMI) for the years 2005-2019 (at a resolution of 0.25° × 0.25°) and the TROPOspheric Monitoring Instrument (TROPOMI) for 2018-2023 (at a resolution of 0.1° × 0.1°). These datasets were collected under partially cloud-free conditions around 13:30 equatorial local standard time. The two products show strong agreement, with TROPOMI-based $PO_3$ being approximately 10% higher than OMI, which is attributed to higher $NO_2$ and HCHO concentrations noted by TROPOMI.

The DNN algorithm ($PO_3$DNN) accounted for more than 96% of the variance in both the test and training datasets derived from observationally-constrained box simulations across various atmospheric composition campaigns, with a slope close to the unity line. The new algorithm improved the representation of $PO_3$ in remote regions compared to the version developed in Souri et al. (2025), due to the inclusion of water vapor and the use of a more robust regression model. We found $PO_3$DNN to be logically responsive to its inputs during various idealized experiments that involved changing light conditions, pollution levels, and water vapor.

Expectedly, our results indicate that $PO_3$ magnitudes and sensitivity maps are primarily influenced by the levels of ozone precursors, non-linearity of ozone chemistry, and photolysis rates. We revisited the accelerated $PO_3$ observed in Souri et al. (2025) across polluted areas, such as major cities and during biomass burning activities in photochemically active environments. Using sensitivity calculations derived from the new algorithm, we investigated the contributors to $PO_3$ seasonality around the globe. We found that photolysis rates were the primary drivers of $PO_3$ seasonality. During darker months, both the magnitude of $PO_3$ and its sensitivity to $NO_2$ and HCHO decrease due to limited light availability to initiate the $RO_X$-$HO_X$ cycle. This critical trend is not represented by the pollution levels alone, highlighting the necessity of including photolysis rates in ozone sensitivity analyses. Fortunately, we can largely constrain these rates using satellite observations. In regions with minimal variability in photolysis rates (such as the tropics), pollution levels became the main driver of $PO_3$ seasonality.

To demonstrate the application of the algorithm for monitoring locally-produced ozone during an extreme event, we compared $PO_3$ magnitudes and sensitivity maps during a rapid heat wave event over the northeastern U.S. to a typical month. We observed a 21% increase in $PO_3$ across the region, primarily attributed to rising background $NO_2$ levels, potentially resulting from enhanced soil $NO_X$ emissions or the photo- or thermal-dissociation of $NO_X$ reservoirs, along with increases in HCHO in VOC-sensitive areas. These results suggest that the rapid calculation capabilities of our product can aid in identifying the causes behind unusual ozone exceedances triggered by weather events, without requiring substantial expertise or computational resources for fine-tuning chemical transport models. However, it is important to note that those models are essential for comprehensively understanding all physiochemical processes responsible for ozone formation and loss, as they provide explicit and process-based details.

The long record of stable observations from OMI allowed us to generate the first-ever maps of $PO_3$ linear trends from 2005 to 2019. While a global analysis of these trend trends certainly requires a follow-up study, we chose two major cities, Tehan (Iran) and Los Angeles (USA), as a showcase because they both fall into the same chemical conditions but experience opposite $NO_2$ trends. These maps revealed a contrasting trend, with Tehran exhibiting negative trends at the city center and positive trends in the surrounding areas. We did not notice significant changes in HCHO, which may be due to the detection limits of OMI. However, mixing ratios of $NO_2$ PBL markedly decreased in Los Angeles and increased in Tehran, a reflection of the implementation and neglect of effective emission mitigation strategies, respectively. As both cities fall into the VOC-sensitive category and are negatively correlated with $NO_2$ concentrations, the differing patterns in their $PO_3$ trends result from the non-linear chemistry of ozone. This



observation was reinforced by the opposite trends in the surrounding areas, suggesting that reductions in
NO$_2$ in Los Angeles significantly lowered ozone formation in the outskirts, while increases in Tehran lead
to higher PO$_3$.
We error characterized both systematic and random errors associated with PO$_3$DNN for both OMI
and TROPOMI-based products. We showed that total errors range from 25% to over 200%, with smaller
errors in polluted areas. Random errors are minor on monthly-basis, with OMI exhibiting larger errors due
to row anomaly issues. Systematic errors exceed 90% of the total error, primarily driven by MINDS
conversion factors. The total errors budget emphasizes on the role of model used for converting satellite-
based VCDs to near-surface concentrations and its importance for precisely determining ozone precursors
levels near to the surface.
We developed a novel product aimed at enhancing our understanding of the variability in PO$_3$ and
its interactions with NO$_X$ and VOCs on a global scale. This advanced algorithm has undergone meticulous
tuning and training using an extensive dataset derived from a reliable box model, which is further
constrained by intensive atmospheric composition campaigns conducted by NASA and NOAA. The
algorithm not only yields accurate estimates of PO$_3$ with minimal bias in comparison to observationally-
constrained values but also facilitates the derivation of PO$_3$ in relation to HCHO and NO$_2$. However, as
indicated by Souri et al. (2025), there remain several opportunities for further improvement, including: i)
the incorporation of heterogeneous chemistry; ii) consideration of the impact of partially cloudy regions
and aerosols on photolysis rates; iii) the inclusion of more sophisticated chemical mechanisms for the
generation of the training dataset; and iv) enhanced representation of vertical profiles of NO$_2$ and HCHO
using observationally-constrained chemical transport models. Some of these enhancements present
significant challenges, particularly the fine-resolution three-dimensional characterization of aerosol and
cloud properties on a global scale, which is not obtainable with current reanalysis data. However, with the
advent of newer satellite technologies such as PACE and MAIA, there may be opportunities to improve the
representation of atmospheric models with respect to cloud and aerosol characteristics.
The emergence of novel geosynchronous orbit (GEO) technologies is becoming increasingly
important for monitoring the daylight hourly variability in ozone precursors. In particular, the finer spatial
and temporal resolution offered by the Tropospheric Emissions: Monitoring of Pollution (TEMPO),
Geostationary Environment Monitoring Spectrometer (GEMS), and Sentinel-4 instruments will aid in
distinguishing exceptional events from typical atmospheric conditions. In light of the success of emission
mitigation strategies over high income countries, the occurrences of elevated PO$_3$ are becoming more
infrequent, thereby necessitating a more detailed and rapid observational strategy for monitoring such
events. This presents a timely opportunity to address ozone exceedance events using TEMPO in conjunction
with our PO$_3$ estimator, especially since the algorithm is designed to handle light-limited conditions—such
as those encountered during early morning and late afternoon periods when TEMPO collects data—
conditions that are not feasible to analyze via the FNR approach.

## Appendix A: The sensitivity maps are the directional derivative

To demonstrate that the sensitivity calculation of PO$_3$ to its inputs resembles (Eq.5) a directional derivative
output, we can approximate the perturbations in the PO$_3$DNN (denoted as $f(x)$, where $x$ is the targeted
sensitivity parameter) using the Taylor expansion:
$$f(1.1x) \approx f(x) + (1.1x - x)\nabla f(x) = f(x) + 0.1x.\nabla f(x) \qquad (12)$$
$$f(0.9x) \approx f(x) + (0.9x - x)\nabla f(x) = f(x) - 0.1x.\nabla f(x) \qquad (13)$$
The sensitivity calculation presented in Eq.3 can be rewritten in the following form:



$$S = \frac{(f(x) + 0.1\nabla f(x)) - (f(x) - 0.1\nabla f(x))}{0.2} = \frac{0.2x.\nabla f(x)}{0.2} = x.\nabla f(x) \qquad (14)$$

Therefore, the first-order approximation of the DNN prediction, when using the given sensitivity
calculation, is $x.\nabla f(x)$ which represents the first-order Taylor expansion term that describes how the output
changes with respect to both the gradient and the magnitude of $x$ (i.e., directional derivative).

## Appendix B. MINDS conversion factor validation

We validate the column conversion factors obtained from the MINDS simulations against
corresponding values derived from aircraft spirals from several suborbital missions. The concentrations of
HCHO and $NO_2$ in both datasets are collocated in time and space and are resampled onto a common vertical
grid, ranging from the near surface up to 450 hPa in 20 hPa increments. To determine the conversion factors,
these resampled concentrations are averaged within the PBL and then divided by the vertically integrated
partial columns from the surface to 450 hPa. The PBLH is based on the MINDS simulations. Figure B.1
displays scatterplots of the paired conversion factor binned at 12:15 LST and 15:15 LST (±45 minutes
around the TROPOMI/OMI local revisit time) for $NO_2$ and HCHO, respectively. The unit for these
conversion factors is ppbv/col, where col represents $1\times10^{15}$ molec.cm$^{-2}$. The comparison shows a good level
of agreement between the two datasets for both species ($R^2>0.7$). The MINDS simulations perform slightly
better for $NO_2$ than for HCHO. This performance difference may arise from the fact that HCHO is mainly
a secondary product, meaning various uncertain VOC emissions, along with uncertain chemical processes
in the model, could pile up leading to discrepancies in the vertical distribution of simulated HCHO
compared to observations. Furthermore, HCHO vertical profiles can be easily affected by local circulation
patterns that are difficult to resolve in coarse models (Souri et al., 2023b). We observe consistent model
performance across various campaigns, except for DISCOVER-AQ Colorado. This discrepancy may result
from complex topography and wind conditions in that region that the model might not fully capture. The
differences between the two datasets can also be attributed to sources of error beyond the model
deficiencies. For instance, the MINDS simulations represent a quarter-degree averaged concentration,
which differs from the localized air samples derived from aircraft, known as the spatial representation error
(Souri et al., 2022).
To account for the systematic errors resulting from the MINDS simulation in our error budget, we
assign $e_{conv\text{-}HCHO}$ and $e_{conv\text{-}NO2}$ in Eq.5 to RMSE values obtained from the comparison. The choice of RMSE
is based on the fact that it contains information about the bias and the dispersion of MINDS with respect to
the observations. We assume these errors to be invariant by time or location, mainly because of limited
aircraft spirals ($N=57$) we have from the suborbital missions.



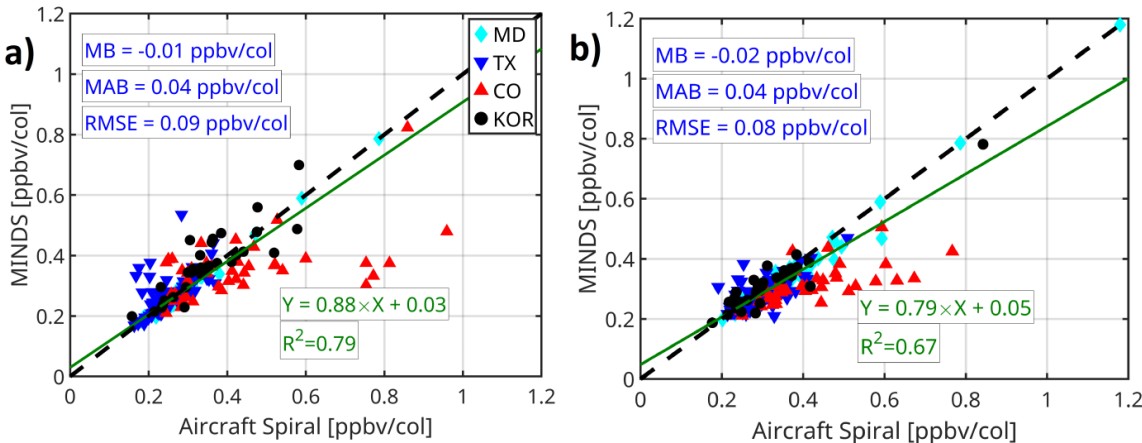

95
**Figure B.1.** The scatterplot of the column to the PBL conversion factor for (a) $NO_2$ and (b) HCHO obtained
from aircraft spirals (x-axis) and MINDS simulation (y-axis) at the same time and location from four
different suborbital missions. These 57 spirals are limited to OMI/TROPOMI overpass ±45 min buffering
time. "col" denotes $1\times10^{15}$ molec.cm$^{-2}$.



**Financial Support**

This study is funded by NASA's ACMAP/Aura project (grant no. 80NSSC23K1250).

**Data Availability**

The $PO_3$ products can be obtained from https://www.ozonerates.space.

TROPOMI satellite data are derived from copernicus Sentinel-5P (processed by ESA), 2021, TROPOMI Level 2 Nitrogen Dioxide total column products. Version 02. European Space Agency. https://doi.org/10.5270/S5P-9bnp8q8, and copernicus Sentinel-5P (processed by ESA), 2020, TROPOMI Level 2 Formaldehyde Total Column products. Version 02. European Space Agency. https://doi.org/10.5270/S5P-vg1i7t0. The TROPOMI UV DLER can be obtained from https://www.temis.nl/surface/albedo/tropomi_ler.php (last access: 10 Nov 2024). OMI SAO HCHO at https://waps.cfa.harvard.edu/sao_atmos/data/omi_hcho/OMI-HCHO-L2/ (last access, 15 Feb 2025). MINDS simulations can be obtained from https://portal.nccs.nasa.gov/datashare/merra2_gmi/gmi-minds/ (last access, 10 April 2025). OMI NO2 (QA4ECV) can be downloaded from https://www.temis.nl/ (last access, 10 April 2025).

**Competing interests**

Bryan N. Duncan is a member of the editorial board of Atmospheric Chemistry and Physics

**Acknowledgements**

Resources supporting this work were provided by the NASA High-End Computing (HEC) Program through the NASA Center for Climate Simulation (NCCS) at Goddard Space Flight Center.

**Authors' contributions**

AHS designed and implemented the research idea, analyzed the data, made all figures except for Figures 3 and 4, and wrote the manuscript. GG implemented, designed, and validated the DNN algorithm, and made Figures 3 and 4. LDM provided the MINDS simulations. BND helped with the interpretation of the results and editing.



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
