# Peer review of "Beyond Binary Maps from HCHO/NO2: A Deep Neural Network"

_EGUsphere, 2025_

## Author Comment (AC1)

This study develops global estimates of ozone production and its sensitivities using satellite observations from OMI and TROPOMI. The method is complicated, which involves box model, CTMs, observations from several field campaigns, synthetic data, satellite data etc. The authors provide a fairly detailed description of the methods, but it remains unclear how these new ozone production estimates advance our understanding of ozone chemistry. My detailed comments are provided below.

**We thank the reviewer for his/her constructive comments, our response follows.**

1. Title: The title begins with 'Beyond HCHO/NO2', which is confusing. What does the term 'Beyond' mean here? How is your study relevant to HCHO/NO2 From the title, one would expect that satellite HCHO/NO2 ratios are central to the analysis, but that does not appear to be the case after reading the manuscript. I'd recommend remove 'Beyond HCHO/NO2'. A novelty of this study (comparing with Souri 2025) is the use of neural network model, and it should be emphasized in the title.

**Response**

We understand that readers may find it difficult to see the connection between FNR and the present study at first sight. To address this ambiguity, we need to provide clearer context and revise the title accordingly.

As stated in the introduction, our work aims to provide two key outputs:

- 1. The magnitude of net PO3: essential for identifying where ozone is locally produced or lost through secondary chemical pathways.
- 2. Sensitivity maps of PO3 to local NO2 (a proxy for reactive nitrogen) and HCHO (a proxy for VOC reactivity), which are critical for guiding emission control strategies.

Traditional data-driven approaches that use satellite observations to diagnose ozone sensitivities to VOCs and NOx have primarily relied on FNR-based segregation of NOx-sensitive, transitional, and VOC-sensitive regimes. These thresholds are derived from various model realizations, and their error structures have been characterized in Souri et al., 2023 and the references therein: https://acp.copernicus.org/articles/23/1963/2023/

**However, FNR has major blind spots:**

1. Lack of sensitivity magnitudes: FNR only classifies regimes without quantifying the actual magnitude of ozone sensitivities. For example, if  $\partial P_{0_3}/\partial NO_2$  is  $+10~\rm s^{-1}$  or  $+3~\rm s^{-1}$ , both would be labeled "NOx-sensitive," even though their regulatory implications might be different. What truly matters about emission control is the magnitude of these responses. For this reason, CTM-based calculations (either through direct decoupled methods, perturbation or adjoint approaches) are typically used. These, however, require extensive efforts to constrain model inputs with satellite data (see Souri et al., 2020:

https://acp.copernicus.org/articles/20/9837/2020/).

Our work provides quantitative first-order sensitivity maps, equivalent to directional derivatives (Appendix A), which is a major innovation of the new algorithm.

2. Lack of adequate dimensions: FNR slices the inherently multidimensional, nonlinear system into just two dimensions. To demonstrate this shortcoming, we perturbed photolysis rates over polluted regions during the KORUS-AQ campaign using observationally-constrained F0AM model. Multiplying photolysis rates by factors of 0.5 (dim, left), 1.0 (default, middle), and 2.0 (bright, right) produced three sets of PO3 isopleths.

The results clearly show that increasing light intensity raises both net PO3 and its sensitivities to NOx and VOC (the contours are more compact in the bright case; each contour corresponds to 3 ppbv/hr). This means that the same FNR can correspond to entirely different magnitude of sensitivities depending on available light. Although one might expect FNR to indirectly reflect variations in photolysis rates, our analysis of 47,000 data points obtained from KORUS-AQ measurements showed no relationship between measured  $jNO_2$  and FNR:

A similar limitation arises from FNR's inability to account for water vapor effects on PO3. Capturing these complex nonlinear interactions between PO3, light, humidity, and precursor concentrations requires more advanced methods over a simple ratio, lacking any information about light intensity and humidity. In a data-driven framework, this is best achieved using nonlinear parameterizations such as DNNs.

This new product therefore represents a paradigm shift away from oversimplified FNR approaches. It not only provides spatiotemporal sensitivity magnitudes, but also accounts for multidimensional dependencies. We highlight this feature in Section 4.3.

For these reasons, we strongly believe this message deserves to be reflected in the title of the paper: it signals a shift toward a more rigorous, multidimensional exploitation of satellite observations for ozone chemistry.

**Modifications**

To better inform how the new sensitivity maps can eliminate the need for FNR and to highlight the machine learning aspect, we added:

"Beyond Binary Maps from HCHO/NO2: A Deep Neural Network Approach to Global Daily Mapping of Net Ozone Production Rates and Sensitivities Constrained by Satellite Observations (2005–2023)"

While we had provided context about the advances made compared to FNR, we added a paragraph in the introduction describing why we should quantify the multidimensional magnitude of PO3 sensitivity, currently lacking in FNR-based approaches. We added in the introduction:

The overarching goal of producing ozone chemistry sensitivity maps is to inform regulatory agencies about the impact of emission reductions on locally produced ozone. Unlike conventional FNR-based binary maps, these maps must quantify the magnitude of sensitivity rather than merely indicating its direction. This quantitative approach is essential because both the sign and magnitude of sensitivities are crucial for understanding the impact of emission changes. While detailed sensitivity maps can be derived from chemical transport models by perturbing underlying emissions, the lack of observational constraints on these models can introduce significant biases. Souri et al. (2025) attempted to address this limitation by providing magnitude-dependent sensitivity maps of PO3 to NO2 and HCHO using piecewise linear regression. However, their approach yielded derivatives of PO3 with respect to NO2 and HCHO that remained invariant with changes in light and humidity conditions. This limitation is problematic because reduced light conditions are known to substantially dampen the sensitivity of PO3 to NOx and VOCs, even under identical emission rates. The current work is therefore motivated by the need to capture the complex, multidimensional dependencies of PO3 on ozone precursors, light intensity, and humidity using a more flexible data-driven approach through a machine learning algorithm. While these maps will not replace process-based chemical transport model experiments, they can efficiently provide first-order assessments to: (i) strategize topdown modeling experiments, (ii) gauge the added value of satellites on predictions of PO3 and (iii) guide the design of sub-orbital missions in regions with poorly documented elevated PO3.

2. For the abstract, the opening should clearly define the scientific question being addressed, rather than starting with the discussion of the FNR, which is not the main focus of this study. My understanding is that this work aims to derive PO3 from a DNN model, which is different from the indicator ratio or FNR approach. The repeated references to FNR throughout the abstract are confusing and should be reconsidered.

**Response**

Our work aims to generate two key products: the net PO3 and the magnitude of PO3 sensitivities to NO2 and HCHO. These two pieces of information are essential for identifying ozone production hotspots and assessing their sensitivity to local pollution levels. This central message should be highlighted in the abstract.

Over the past two decades, we have extensively explored the application of FNR in diagnosing ozone chemistry (e.g., Duncan et al., 2010; Souri et al., 2020; Souri et al., 2023). While FNR has been a valuable first step in demonstrating the utility of satellite observations to classify ozone chemical regimes, it ultimately offers only a binary perspective on a fundamentally continuous and multidimensional problem. Therefore, it is essential to highlight this new fresh paradigm.

**Modifications**

In the supplementary, we added a new section describing the fundamental issues with FNR; we did not include it in the main draft because it is more of a reminder for people who may misuse FNR rather than bringing new insights into ozone chemistry.

**1. FNR is oblivious to the impact of photolysis rates and water vapor content on PO3**

The primary objective of using the formaldehyde-to-nitrogen dioxide ratio (FNR) is to reduce high-dimensional, non-linear ozone production rates into a two-dimensional framework based on volatile organic compound reactivity (VOCR) and reactive nitrogen. However, beyond the fact that HCHO and NO2 does not fully represent VOCR and reactive nitrogen, it is crucial to recognize that ozone production rate sensitivities and magnitudes depend on other geophysical variables independent of FNR. Among these variables, photolysis rates and water vapor are major drivers of atmospheric oxidation capacity, modulating numerous reactions related to ozone production (Kleinman et al., 2001).

To demonstrate photolysis rate effects on both PO3 magnitudes and sensitivities, we conducted F0AM box model simulations constrained by geophysical variables during June 6-9 of the KORUS-AQ campaign (Souri et al., 2025). We perturbed NOx, VOCs, and photolysis rates to generate three sets of isopleths (Figure S1). The results clearly show larger ozone production rates under more intense light conditions. More importantly, the contours corresponding to identical PO3 intervals (3 ppbv/hr) become more compact under brighter conditions, indicating that PO3 becomes more sensitive to both NOX and VOCs with increased light intensity. This pattern suggests that identical FNR values under different photolysis rates can have fundamentally different implications for ozone production rate sensitivities.

To confirm that FNR contains no photolysis rate information, we analyze paired FNR and jNO2 photolysis rate measurements from over 47,000 data points during the KORUS-AQ campaign, revealing no correlation between these variables (Figure S2). This demonstrates the need for additional dimensions in ozone sensitivity analysis, necessitating more sophisticated algorithms (like our approach) over traditional threshold-based methods.

**Figure S1.** The PO3 isopleths generated using F0AM box models derived from observations taken during the KORUS-AQ campaign under three different photolysis rates scenarios: (left) multiplied by 0.5, (middle) default, (right) multiplied by 2.0. Each contour represents 3 ppbv/hr.

**Figure S2.** The comparison of measured FNR and measured jNO2 frequencies taken from aircraft observations during the KORUS-AQ campaigns. All measured points are used to make this plot.

Figure S3 illustrates the representation of ozone sensitivities by mapping five variables derived from TROPOMI and our PO3DNN parameterization across two seasons over Los Angeles.

FNR values are low during colder months due to abundant NO2 relative to HCHO, qualitatively suggesting the LA region should be predominantly VOC-sensitive. However, the derivatives and sensitivities of PO3 to both HCHO and NO2 remain muted due to limited photochemical activity, making PO3 unresponsive to NOX and VOC concentrations. Conversely, summer conditions yield larger derivatives, showing much stronger PO3 responses to both species. This example can be extended to different times of day, such as FNR values from geostationary satellites or morning versus afternoon measurements from low Earth orbit satellites.

**Figure S3.** Five variables derived from our PO3DNN product based on TROPOMI dataset. The first row focuses on December-January-February (DJF), while the second row shows those variables for June-July-August 2023. The calculation of the sensitivities and derivatives are based on perturbation of the DNN algorithm described in the main paper.

The absence of PO3-relevant geophysical information in FNR also applies to water vapor. F0AM box simulations over polluted regions show that increasing humidity enhances PO3 through the generation of two OH molecules via H2O+O1D reactions (Figure S4). However, FNR contains no water vapor information, as humidity is driven by hydrological and meteorological factors decoupled from the processes determining FNR (Figure S5). This further necessitates adding water vapor as an additional dimension in ozone sensitivity

**Figure S4**. The effect of  $H_2O(v)$  on  $PO_3$  during KORUS-AQ campaigns. Only highly polluted regions (HCHO×NO2 > 10) are selected for this experiment.

**Figure S5.** The comparison of measured FNR and measured water vapor density taken from aircraft observations during the KORUS-AQ campaigns. All measured points are used to make this plot.

3. This study appears to be a follow-up study of Souri et al. 2025 with some technical improvements, such as use of DNN. While the technical enhancements are clear, the added scientific value is not. It is unclear how the improved PO3 estimates advance our understanding of ozone formation processes. Many figures, including the spatial maps and seasonal variations, are quite similar to those presented in Souri et al. (2025). The main difference seems to be the extension of the study period from one year to multiple years (2005–2023), but only two regional case studies are analyzed for long-term trends. I suggest expanding the long-term trend analysis globally to better demonstrate the added value of this extended dataset.

**Response**

Thanks for the suggestion about expanding the trend analysis globally. While we recognize that our previous work has similarities with respect to PO3 predictions compared to the current work, there are distinct differences which are documented in in the paper (improved prediction, more cohesive between remote and polluted regions, substantially reduced noise, and less discretization). In fact, it is encouraging to see that both algorithms provided consistent results on average. The most innovative part of the current approach lies in its ability to provide a more comprehensive sensitivity maps compared to Souri et al., 2025.

We decided to add a global trend analysis (2005-2019) of PO3 with respect to NO2 and HCHO using OMI in the manuscript. We do not intend to include TROPOMI in the long-term analysis because it will require a data harmonization approach which is still under investigation within our team (the objective of the third year of our ACMAP-Aura project). In addition, the long-term stability of OMI radiance has made it a great product to study trend.

**Modifications**

We moved the trend analysis of Tehran and LA to the supplementary material, and replaced that with a global analysis.

We added these global findings to the abstract, introduction and conclusion. In the abstract:

The stability and long-term records of OMI retrievals (2005-2019) enable us to provide the first global maps of PO3 linear trends showing a surge of >20% over China, the Middle East, and India, while a reduction in the eastern U.S., southern Europe, and several regions in Africa.

In the conclusion:

The long record of stable observations from OMI allowed us to generate the first-ever maps of PO3 linear trends from 2005 to 2019 globally. The global long-term trends revealed substantial spatial variability, with predominantly positive trends over Asia and the Middle East (>30% relative to 2005 in some regions) and negative trends across the eastern U.S., Europe, and parts of Africa. Analysis indicated that simultaneous changes in HCHO and NO2 boundary layer concentrations were the primary drivers of these trends. Although increases in both precursors over Asia and the Middle East, rising PO3 and reduced concentrations elsewhere lead to decreases, localized non-linearities complicated this relationship, as demonstrated by contrasting chemical regimes in Tehran vs. Los Angeles. Quantitative attribution of these trends presents significant challenges because of their small amplitudes relative to seasonal variations and non-linear sensitivities in the parameterization, necessitating "hold-one-out" approaches that account for complex interdependencies between input variables.

**4.4.3. Global PO3 linear trends using OMI (2005-2019)**

Using the linear trend calculation method outlined by Souri et al. (2024), we compute global long-term linear trends of PO3 from OMI data, shown in Figure 8. High-latitude regions (>65°) are excluded due to limited photochemical activity. We observe large variability in both the signs and magnitudes of the linear trends. Predominantly positive trends occur over the Middle East, India, and China, while negative trends are mostly found in the eastern U.S., maritime Southeast Asia, and several areas in Africa. The largest upward trend in PO3 over the U.S. occurs in oil and gas producing regions, including the Permian Basin. While various physicochemical processes beyond near-surface PO3 influence tropospheric ozone trends, the strong agreement between predominantly upward PO3 trends in Asia and the Middle East and satellite-based ozone observations (Gaudel et al., 2018; Boynar et al., 2025) is noteworthy.

To gather a more relative perspective, Figure 9 shows relative PO3 trends (as percentages relative to 2005 annual averages) for regions where PO3 exceeds 0.5 ppbv/hr. The largest relative changes (>30%) are evident over the Persian Gulf, Chile, India, and China. Large negative values dominate over the eastern U.S. and over the central Africa (>20%).

Multiple factors in our parameterization can simultaneously influence these trends, including changes in HCHO VCDs, NO2 VCDs, dynamic changes in column-to-PBL conversion factors from MINDS, water vapor, and photolysis rates. However, photolysis rate trends should be negligible because long-term changes in total overhead ozone are insignificant at midlatitudes (Souri et al., 2024), and surface albedo is based on a monthly climatology dataset. While water vapor increases over time in response to global warming (Souri et al., 2024; Borger et al., 2024), these changes are insufficient to explain the large variability in PO3 linear trends over polluted regions. Accordingly, simultaneous changes in HCHO and NO2 boundary layer mixing ratios are the main drivers of PO3 trends.

The PO3 trends are generally explained by changes in ozone precursor concentrations which are mapped in Figures S10 and S11. The attribution of trends in OMI HCHO and NO2 have been partly discussed in Souri et al., 2024 and the references therein. Increases in both HCHO and NO2 over the Middle East, India, and China drive rising PO3 over time. Conversely, reduced HCHO and NO2 concentrations over parts of Africa, the eastern U.S., and maritime Southeast Asia, have led to PO3 reductions. However, many localized areas exhibit strong non-linearity. For instance, Tehran (Iran) shows positive PO3 trends (Figure S13) caused by NO2 increases in a predominantly VOC-sensitive regime, reducing ozone loss through NO2+OH

reactions. Los Angeles (USA) shows upward trends attributed to rapid NO2 reductions, resulting in the opposite effect (Figure S14).

The quantitative characterization of these trends (similar to our analysis of PO3 seasonality in Section 4.4.2 or rapid PO3 changes during a heatwave in Text S2) presents significant challenges for several reasons: (i) the amplitudes of these trends are generally an order of magnitude smaller than seasonal changes, requiring more stringent attribution methods, (ii) the sensitivities of PO3 to input parameterization can behave non-linearly, making a linear trend analysis ill-suited for some localized areas, and (iii) changes in ozone precursors have effects on the sensitivity of PO3 to photolysis rates as described in Section 4.4.2, introducing a convoluted problem.

Since our PO3 parameterization encapsulates non-linear and interdependent relationships between pollution levels, light intensity, and water vapor, fully isolating individual effects on PO3 trends requires reproducing the product while holding either NO2 or HCHO constant individually and allowing others to evolve over time (an approach similar to modeling experiments in Souri et al., 2024). This approach comprehensively captures the non-linear dependencies between input variables and PO3, circumventing the need for crude linear approximations.

**Figure 8.** The linear trend maps of PO3 within PBL derived from our new algorithm using OMI in 2005-2019. Dots indicate that the trend has passed the Mann–Kendall test at 95% confidence interval.

Figure 9. Similar to Figure 8 but percentage changes are instead shown over PO3>0.5 ppbv/hr.

4. It is unclear to me why a satellite-based PO3 product is needed. PO3 is essentially a "modeled" quantity, which is not directly observable. There is no way to evaluate the robustness of PO3 estimates. The magnitude of PO3 can vary depending on how you define the PO3, whether it's accumulative production or instantaneous production. It seems that the authors are looking into net production of O3, but it is not clear how the chemical loss of O3 is defined, and how the uncertainties of chemical loss terms would influence the magnitude of PO3.

**Response**

We respectfully disagree with this comment.

PO3 is not purely a modeling quantity but is measurable using specialized dual-tube instruments (Cazorla and Brune, 2010; Sadanaga et al., 2017; Sklaveniti et al., 2018), as mentioned in our introduction. These instruments can provide valuable insights into chemistry representation in models. While measurement uncertainties are decreasing over time, these instruments remain in the development stage, and we believe our product could help accelerate their improvement and deployment.

We carefully considered how to define PO3 to enable seamless intercomparison with future PO3 estimates. Previously, we examined individual reaction rates defining both production and loss terms (e.g., Souri et al., 2020). However, explicitly defining these terms creates challenges for direct comparison across different chemical mechanisms. For instance, peroxy radicals (RO2) are defined differently among various chemical mechanisms, and some VOC and organic nitrate definitions are inconsistent (some mechanisms use lumped species while others separate them).

A practical approach for defining PO3 in this context is to calculate the instantaneous PO3 tendency by summing all chemical loss pathways of ozone (negative stoichiometric

coefficients) and all chemical production pathways (positive stoichiometric coefficients). This approach closely matches the output from chemical solvers in atmospheric models under steady-state conditions and facilitates intercomparison procedures. While we lose some chemical interpretation regarding individual chemical terms shaping PO3, our product focuses on net values rather than parameterizing individual terms.

We acknowledge that we cannot directly validate F0AM PO3 against measurements due to the absence of PO3 observations during the suborbital missions. However, PO3 is influenced by numerous geophysical variables that are either directly or indirectly constrained in our box model (Section 4.1 in

https://acp.copernicus.org/articles/25/2061/2025/). Examination of individual terms defining PO3 in the CB06 mechanism shows that nearly all are well-constrained in our simulations: we accurately reproduced NO and NO2 compared to aircraft measurements, constrained many VOCs yielding reasonable HCHO simulations against observations, and reproduced HO2 and OH with minimal biases and high correspondence within instrument noise levels. The first-order approximation of PO3 in urban settings (NO+HO2 minus NO2+OH) involves species that are all well-captured in our model.

The primary uncertainty lies in RO2, which serves as a proxy, highlighting where specialized PO3 instruments could help validate constrained PO3 estimates across different chemical mechanisms and heterogeneous chemistry treatments. While we do not claim complete alignment with actual PO3 values (which cannot be verified due to absent measurements), we believe our box model simulations provide reasonable constraints on the various terms contributing to PO3.

**Modifications**

We improved the wording around the PO3 definition in the methodology:

Once the simulations are done, we determine simulated PO3 by:

$$PO_3 = FO_3 - LO_3 \tag{1}$$

where LO3 is all possible chemical loss pathways of ozone (negative stoichiometric multiplier matrix) and FO3 is all possible chemical pathways producing ozone molecules (positive stoichiometric multiplier matrix). This equation is also known as ozone tendency. This definition simplifies intercomparison with estimates derived from different chemical mechanisms by eliminating the requirement to explicitly match individual production and loss terms, which often exhibit inconsistencies across mechanisms, especially in their treatment of peroxy radicals. The calculation of PO3 is under a steady-state assumption.

5. The authors claim that photolysis rates and water vapor have large influence on PO3. However, their calculations of these quantities appear oversimplified. It is unclear how cloud and aerosol effects on photolysis are accounted for. Water vapor and total ozone columns are taken from MINDS simulations, even though satellite-based observations for

these variables are available. It is not clear why satellite data are only used for NO2 and HCHO but not for other relevant parameters. This inconsistency needs to be addressed.

**Response**

No single satellite can reliably measure near-surface water vapor (H2O(v)) at the spatial coverage provided by TROPOMI and OMI. Available satellite capabilities vary significantly: some measure only total column water vapor (MODIS, OMI, TROPOMI), others provide vertical profiles with limited near-surface sensitivity (IASI, AIRS), and GPS radio occultation provides sparse but accurate profiles. The diversity of surface-based, sounding, and satellite instruments for water vapor retrieval, each with unique strengths and limitations, has motivated efforts to integrate them within harmonized frameworks through data assimilation. This approach provides optimal H2O estimates by accounting for varying vertical sensitivity, spatial representation, and sensor-specific artifacts and errors.

We leveraged the well-established MERRA-2 "replay" data assimilation framework, which constrains water vapor using numerous observational products. Our validation against SSMIS integrated water vapor (IWV) (recognized as the most robust water vapor product over oceans, which comprise 71% of Earth's surface) shows minimal biases in 2005 with replay mode enabled in a GEOS-simulation performed in Souri et al. (2024) (figure below).

Our sensitivity analysis reveals that PO3 responses to H2O variations are generally an order of magnitude smaller than those for photolysis rates (Js), NO2, and HCHO, typically ranging around 1-2 ppbv/hr per unit of water vapor density. Therefore, having 1-5% uncertainties in simulated water vapor should not significantly impact our results and would remain even smaller than DNN estimator errors.

Likewise, total ozone columns are constrained by satellites in MINDS with only 2-3% error (see Figure S1 in <a href="https://acp.copernicus.org/articles/24/8677/2024/acp-24-8677-2024-supplement.pdf">https://acp.copernicus.org/articles/24/8677/2024/acp-24-8677-2024-supplement.pdf</a>). Their errors can be safely ignored.

Regarding the impact of aerosols and clouds on photolysis rates, we agree that they can partly introduce biases in our estimates, as discussed in the paper. This error has been largely mitigated by removing clouds/aerosol using the effective cloud fraction being sensitive to all those particles.

There are known physical models to scale photolysis rates given the optical properties of particles (such as FAST-JX or RACM). However, it is not feasible to source 3D optical properties of aerosols and clouds at the same resolution and time as of TROPOMI and OMI globally. While some instruments like TROPOMI can provide 2D optical properties, we are required to know how much of these are below PBL and how much are above it. There are also complexities about the height of aerosols, because aerosol layer height from TROPOMI or OMI is optical centroid and not the physical top boundary. Knowing these optical properties (partial AOD, SSA, and phase functions) is essential.

Similar to the discussion about water vapor, we need a data assimilation approach to exploit various ground and space remote sensing instruments to constrain aerosols and cloud optical properties in models. But this is much more challenging compared to the water vapor problem, because aerosols and clouds are affected by a larger number of physiochemical processes. While we could have used MINDS cloud/aerosol optical properties to supposedly scale photolysis rates, we think the errors and mismatches of the model would have harmed the analysis.

We also need to emphasize that the effective cloud fraction is not equal to geometrical cloud fraction (defined in meteorology). The O2-O2 algorithm is sensitive to the amount of contamination by clouds (even over sensitive to thin clouds), making the cloud flag a effective to mask them. To show some showcases for our daily OMI PO3 product:

This is another TROPOMI case that shows in strong smoky areas in California, the quality flags removed most of the contaminated pixels (but not all).

**Modifications**

To address this comment, we moved the discussion about the effect of clouds and aerosol in Stage 1 to the error analysis part and added more caveats:

It is important to acknowledge that the defined total error budget here is only a good guess and optimistic. Some underlying sources of error, which are difficult to quantify, are not included. For example, errors related to the training dataset derived from the F0AM model are challenging to assess because of the lack of PO3 measurements. We assume other inputs to the PO3 parametrization, such as the monthly climatology TROPOMI surface albedo to be errorfree. Additionally, all datasets used to estimate PO3 contain spatial representation errors (Souri et al. 2023), which are difficult to measure without knowing their true state of global spatial variability. It is worth noting that some of the inputs such as H2O(v) and the overhead ozone column have minimal biases because of MINDS simulations being observationally constrained (Fisher et al., 2024; Souri et al., 2024).

Another source of uncertainty arises from partially cloudy pixels and aerosols, which can introduce errors in calculated photolysis rates. While we successfully filtered out cloud cover and strong aerosol loadings (e.g., from wildfires) using effective cloud fraction thresholds, some aerosol or cloud-contaminated pixels may pass cloud screening due to low optical depth or height characteristics. Rigorously quantifying the errors coming from these effects would require running a radiative transfer model with detailed three-dimensional optical properties of

clouds and aerosols on a global scale, particularly critical for aerosols, which can have complex effects on photolysis rates depending on their absorption and scattering properties and vertical distribution. Unfortunately, such comprehensive datasets are typically limited to the narrow swaths of spaceborne lidar observations, which themselves carry substantial uncertainties (Thorsen and Fu, 2015). While these complications cannot be entirely avoided, particularly for aerosol effects, users can apply additional quality control measures by filtering pixels using aerosol optical depth retrievals from TROPOMI, OMI, or other sensors to more rigorously identify contaminated observations.

6. The authors demonstrate the use of PO3 through some case studies, but these studies are somewhat disconnected. Each focuses on a different region and time period (e.g., northeastern U.S., Middle East, Los Angeles, Tehran), resulting in a fragmented narrative that feels like a collection of isolated examples. I recommend reorganizing these sections to tell a more cohesive scientific story. The analysis of long-term trends is promising. Expanding this analysis to the global scale, and examining how ozone production sensitivities have evolved over time, would substantially strengthen the manuscript.

**Response**

We agree that these are different applications which were meant to provide more confidence in the utility of our product from different angles. We reordered some of the sections and moved some to the supplementary materials to have a more cohesive flow.

**Modifications**

We renamed Section 4.4:

PO3 Maps and Sensitivities using OMI and TROPOMI: A General View, Long-term analysis, and Intercomparisons

Now this section starts with 4.4.1. Global PO3 and Seasonality using OMI in 2005-2007 The reason behind it is that 2005-2007 is when OMI signal was strong and did not go through the row anomaly issues. We then have their attributions in 4.4.2. The attribution of PO3 seasonality.

We then introduced "4.4.3. Global PO3 linear trends using OMI (2005-2019)" to keep the discussion focused on OMI.

Then we introduce TROPOMI and its intercomparison with OMI. This is good bridge to move from OMI to TROPOMI while having some joint discussion: 4.4.4. High resolution TROPOMI-based PO3 maps contrasted with OMI in 2019

Then we have 4.4.5. Error Analysis to discuss both OMI and TROPOMI errors on a monthly basis.

Finally, we have this section separating the sensitivity map analysis from the rest: 4.4.6. Beyond binary maps: Ozone sensitivity maps using high-resolution TROPOMI data

As a result, the discussion about LA and Tehran and the heatwave effect have been moved to the supplementary. We think the new layout is more cohesive than before.

7. The DNN model is trained using F0AM-simulated data. Although the model shows reasonable performance, the derived relationships remain model-dependent and limited by the diversity of available field campaigns. Rather than randomly withholding data for testing, it would be more informative to exclude one or two entire field campaigns from training and test whether the DNN performs well out-of-sample. This approach would better demonstrate the model's robustness and generalizability.

**Response**

Thanks for the suggestion! We performed the similar experiment as the reviewer suggested for PO3LASSO in Souri et al., 2025, but we decided to show "test" data as they were never used for hyperparameter tuning. We added this new figure in the supplementary with the campaign-specific withholding figure, compare to Figure 7 in Souri et al., 2025.

**Modifications**

**We added:**

Similar to the approach of Souri et al. (2025), we completely exclude each suborbital mission from the training dataset and use it as an independent benchmark to evaluate the model's performance. The resulting accuracy is comparable to that achieved when 56% of the data are used for training, indicating that the PO3 parameterization has reached a high degree of generalization (Figure S10).

**Figure S10.** Each campaign dropped from training PO3DNN and subsequently used as an independent benchmark.

Specific comments:

1. Line 155: Unclear what the offset and slope mean.

**Response**

Corrected.

**Modifications**

To correct for offset (additive bias) and slope (multiplicative bias) in this product

2. Line 167: Why different cloud fraction thresholds are applied to NO2 vs. HCHO.

**Response**

We strictly used the recommended values based on their user guide or commonly-used thresholds. However, it is important to note that, because PO3 is produced on daily basis from both HCHO and NO2, a stricter flag between these products dictate where we should discard the unfit pixels. For instance, if ECF threshold is set to 10% for NO2, but 90% for HCHO, the 10% becomes the determining factor. As shown in this response letter, we don't think clouds will be a major problem in our analysis.

3. Line 401: The assumption stated here seems questionable. MINDS-simulated water vapor and photolysis rates carry uncertainties, the influence of clouds and aerosols is not accounted for. These sources of uncertainty should be incorporated into the error analysis.

**Response**

We addressed this in the reviewer's major comment.

It is not straightforward to characterize the errors in photolysis rates without precisely knowing 3D optical properties of clouds/aerosols and surface albedo reflectivity. While we could have thrown some numbers to propagate the errors, we think the quality of error characterization should be on par with the rest of the analysis.

4. Figure 6: While the absolute PO3 values vary between bright and dim conditions, the spatial patterns (e.g., the ridgeline) appear consistent? It would be helpful to label the ridgeline across all panels.

**Response**

While this is a valid point, we are against binarization of the atmospheric conditions. Having more red tapes on these contours will indirectly encourage people to see only the sign of the sensitivities, however, as stated in our work, we should consider the magnitude of the sensitivities to better describe ozone responses to its precursors.

5. Figure 8: I'm having a hard time interpreting the sensitivity terms. What exactly do these sensitivities represent? Given that the magnitudes of photolysis rate, HCHO, and NO2 differ substantially, and that ozone chemistry is highly nonlinear, are these sensitivities additive?

**Response**

We had provided the mathematical meaning of these sensitivities in Appendix A. They are the directional derivative providing the first-order sensitivity.

If we sum them, using a Taylor expansion, they will explain the first order approximation of PO3 minus a constant value. However, as the reviewer stated, PO3 is a non-linear problem and so is the DNN. So in order to better approximate PO3, we should also calculate higher order derivatives. We did not provide second-order sensitivities (which can be calculated in this way:  $S^{(2)} = [C(+\Delta\epsilon) - 2C(0) + C(-\Delta\epsilon)]/(\Delta\epsilon)^2$ ), but we think the first-order sensitivities are adequate to describe the seasonality of PO3. Basically, the sum of these three terms explain most of the amplitude of the seasonality minus a constant offset.

6. Figure 8: The higher sensitivity of PO3 to HCHO in summer does not necessarily imply stronger sensitivity to VOC emissions. This may simply reflect the shared temperature dependence of PO3 and HCHO. In CTMs, ozone sensitivity is typically analyzed with respect to VOC emissions, whereas HCHO is an intermediate oxidation product rather than a primary species. The production of HCHO varies with VOC speciation, NOx levels and temperature.

**Response**

This is a valid point, which is why we carefully specify that these sensitivities relate PO3 to HCHO and NO2 concentrations rather than emissions. The observed HCHO and NO2 concentrations reflect the integrated effects of emissions, meteorology, transport, deposition, and chemistry. Our approach captures these combined processes within the product, though we cannot separate their individual contributions.

**Modifications**

**To reemphasize it we added:**

Photolysis rates, which serve as crucial indicators of photochemical activity, are the primary determinants of PO3 seasonality. Figure 8 illustrates the sensitivity of PO3 to NO2, HCHO, and

combined J-values (jNO2 and jO1D) based on Eq.3 across the same regions and months presented in Figure 7. The absolute values of PBL HCHO, NO2, and jNO2 are shown in Figure S3. As shown in Appendix A, these sensitivity values are influenced by both the magnitude of the precursor and the first derivative of PO3 with respect to that precursor. Thus, the sensitivity values should be interpreted as the result of these combined effects. Moreover, these sensitivities are calculated with respect to local HCHO and NO2 concentrations rather than local emissions (unlike typical modeling experiments). Local concentrations reflect the combined influence of both local and external emissions through various physicochemical processes.

---

## Author Comment (AC2)

The manuscript presents a new global dataset of net ozone production rates (PO3) derived from a neural network framework constrained by satellite observations. The authors develop a deep neural network (DNN) model trained with simulations from the F0AM box model and aircraft measurements with perturbations. The DNN employs as input a set of geophysical parameters derived from multiple sources, including satellite retrievals of HCHO and NO2 (from OMI for 2005–2019 and TROPOMI for 2018–2023), as well as parameters from the MINDS model. The MINDS framework provides the conversion factors from total column to planetary boundary layer (PBL) mixing ratios for HCHO and NO2, simulated O3, and water vapor (H2O).

The authors validate their DNN-derived PO3 product (termed PO3DNN) against the empirical formulation described in Souri et al. (2025). The manuscript further examines several applications: (i) the intercomparison of OMI- and TROPOMI-based products for 2019, (ii) regional PO3 seasonality (2005–2007) across selected sites worldwide, (iii) a heatwave case study over the northeastern United States (August 2007), (iv) seasonal and spatial patterns over the Middle East (2019), and (v) long-term PO3 trends from 2005–2019 in Los Angeles and Tehran.

Comprehensive uncertainty estimates are presented, including both systematic and random errors, with the dominant source attributed to the column-to-PBL conversion factors from MINDS. The total relative errors are reported to range from about 25% in polluted regions to more than 200% in remote areas.

The manuscript is scientifically interesting and presents a valuable global dataset of ozone production rates derived from satellite observations. However, several key methodological and interpretational issues need to be clarified and better quantified before the study can be fully evaluated. Therefore, I consider the following as major comments.

**We thank the reviewer for his/her constructive comments and detailed summary, our response follows.**

Major Comments

1. Time period, harmonization, and trend (2005–2023)

The title suggests that the study spans the full period of 2005–2023, implying a continuous long-term trend analysis that combines OMI and TROPOMI data. However, the actual trend analysis uses only OMI data (2005–2019), while TROPOMI is primarily used for 2019 onward and inter-satellite comparison. The manuscript should clarify how the two products were harmonized for consistency and provide quantitative evidence of their agreement or bias (for example, regional mean differences, temporal overlap). It would also strengthen the study to explicitly show the magnitude and spatial or temporal characteristics of any applied corrections and to quantify how harmonization affects the derived PO3 trends (for example, OMI-only vs. TROPOMI-only vs. combined). Finally,

please discuss whether a unified 2005–2023 trend is feasible and what systematic offsets might influence its interpretation.

**Response**

We acknowledge that the title may create confusion about whether both products (TROPOMI and OMI) have been fully harmonized for long-term trend applications.

Harmonization is inherently subjective and depends on user-defined tolerance levels. While our products demonstrate consistency within 10% during a joint year (2019), whether this level of agreement is sufficient for robust trend calculations depends on both the user's tolerance requirements and the magnitude of the trends being analyzed.

Both products use identical models for conversion factors, photolysis rates, water vapor calculations, and a forward DNN estimator. The primary differences come from variations in NO2 and HCHO VCDs.

Complete harmonization requires consistent retrieval algorithms across both TROPOMI and OMI, including:

- Identical RTMs with consistent assumptions for O2-O2 algorithms, surface properties, ocean properties, and so forth.
- Uniform slant column fitting approaches (using identical spectral windows, cross-sections, and fitting methods; whether DOAS or BOAS)
- Consistent a priori assumptions

This needs a substantial undertaking. Well-established, long-term projects such as NASA's MEaSUREs and QA4ECV have been developed specifically to create consistent algorithms for robust long-term analysis. To our knowledge, such comprehensive harmonization has not yet been achieved for TROPOMI, and our current budget constraints prevent us from harmonizing the satellite VCDs to this extent.

We have proactively implemented robust bias correction against ground-based remote sensing observations, which is a critical component of data harmonization. This approach prevents both satellites from diverging significantly from established benchmarks as a first-order approximation, resulting in our products' 10% agreement (Figure\*). Whether this makes up sufficient "data harmonization" ultimately depends on users' requirements for trend derivation robustness.

A harmonization effort should be part of a "post-processing" step. Even if we had implemented additional harmonization approaches for deriving 2005-2023 trends (an objective of our final ACMAP-Aura project year), users would still need to apply their preferred harmonization method, as no single standardized approach exists.

An important factor in trend analysis is that TROPOMI's long-term stability for HCHO and NO2 measurements has not been as thoroughly validated as OMI's record. Recent studies have documented potential drifts in TROPOMI data products that warrant further investigation (<a href="https://amt.copernicus.org/articles/17/3969/2024/">https://amt.copernicus.org/articles/17/3969/2024/</a>).

In summary, we are unable to fully harmonize TROPOMI and OMI NO2 and HCHO retrieval algorithms to create a consistent PO3 product. While the application of the bias correction using MAX-DOAS/FTIR has made both products agree well within 10%, we think the users may need to apply a harmonization algorithm as a post-processing step. We need to provide this caveat and limitation in the summary section.

**Modifications**

We added this paragraph to the summary:

While the OMI- and TROPOMI-based PO3 products maintain algorithmic consistency in several key components, including photolysis rates and water vapor, the underlying satellite retrievals of HCHO and NO2 VCDs remain unharmonized between the two instruments. To address the resulting inter-instrument biases, we implemented bias correction using ground-based remote sensing retrievals as reference standards. This approach achieved OMI and TROPOMI PO3 agreement within 10% on average. However, this level of consistency may be insufficient for robust joint trend analysis of the combined OMI-TROPOMI PO3 record over areas with non-linear or small trends, potentially requiring the implementation of trend harmonization algorithms (e.g., Hilboll et al., 2013) to warrant statistical reliability in long-term analyses.

2. Bias correction using MAX-DOAS

The bias correction procedure for OMI and TROPOMI retrievals using MAX-DOAS

observations needs more detail. Were corrections applied globally or regionally, and are
they time dependent? How large were the typical corrections? The error treatment also
appears simplified for bias correction and may underestimate correlated uncertainties.

**Response**

The reviewer is right about the fact we did not account for correlated uncertainties among HCHO and NO2 VCDs in the error budget or the linear fit between satellites vs. benchmarks (although the errors in x and y are considered in Souri et al. 2025 based on weighted chi2 minimization).

TROPOMI HCHO bias correction comes from Souri et al. (2025) who expanded the analysis of Vigrouroux et al. (2021) using FTIR measurements. A similar work was recently done with OMI HCHO with the same dataset (Ayazpour et al., 2025) whose correction factors were used in our work. OMI NO2 correction factors were derived from a established work done by Pinardi et al. (2021), and TROPOMI NO2 follows the work we did in Souri et al. (2025) comparing MAX-DOAS and the satellite observations based on an extension to Verhoelst et al. (2020).

We added a new section right after introducing TROPOMI and OMI retrievals to elaborate on the number of stations, duration, and the magnitude of these corrections.

The ground remote sensing data used are global.

The reviewer raised concerns about the generalizability of our benchmarks, mentioning their sparse distribution and potential for varying satellite discrepancies across seasons and locations. However, numerous validation studies (Verhoelst et al., 2020; Vigouroux et al., 2021; Ayazpour et al., 2025; and Table 1 in

https://acp.copernicus.org/articles/21/18227/2021/) have demonstrated that biases in TROPOMI and OMI NO2 and HCHO columns consist of two components: an additive term (offset that exists uniformly regardless of season or location) and a multiplicative term (magnitude-dependent slope).

The rationale for parameterizing retrieval biases as a function of magnitude is to enhance correction factor generalizability across seasons/locations. By understanding how bias changes with magnitude, we can predict seasonal bias variations since column densities vary seasonally.

A remaining question is whether these slopes and offsets remain consistent across different locations and seasons? This is precisely why we included bias correction error terms in our analysis. If the relationship between benchmarks and satellite retrievals varied dramatically by location or season, linear fits would become highly uncertain, resulting in large coefficient errors.

Figure 8 and the validation studies referenced above

(https://acp.copernicus.org/articles/25/2061/2025/acp-25-2061-2025.pdf) suggest the opposite, indicating that these parameters are mostly reliable. Similarly, if we had insufficient samples, this would have resulted in larger uncertainties associated with the slopes and offsets.

Lastly, would adding a new instrument over a different area change the correction factors? This question would fall into the area of "unknown unknown". We won't know until we measure. The current correction factors applied are based on the most recent and credible validation efforts (the known known).

**Modifications**

We clarified that the correlated errors aren't considered in the error characterization:

It is important to acknowledge that the defined total error budget here is only a good guess and optimistic. Some underlying sources of error, which are difficult to quantify, are not included. For example, errors related to the training dataset derived from the F0AM model are challenging to assess because of the lack of PO3 measurements. We assume other inputs to the PO3 parametrization, such as the monthly climatology TROPOMI surface albedo to be error-free. Additionally, all datasets used to estimate PO3 contain spatial representation errors (Souri et al. 2023), which are difficult to measure without knowing their true state of global spatial variability. Moreover, we do not consider correlated errors among HCHO and NO2 retrievals.

We added a new section right after defining the TROPOMI and OMI datasets:

**1.1.1. Bias correction using ground-based remote sensing data**

In order to remove large biases in both TROPOMI and OMI products, we bias correct their columns using the offset (additive term) and slope (multiplicative term) determined from a linear fit to paired MAX-DOAS/FTIR and these datasets, as described by Souri et al. (2025). The rationale for defining retrieval biases as a function of magnitude is to enhance correction factor generalizability across seasons and locations. We take advantage of three studies characterizing the bias correction factors, listed in Table 1. The application of these correction factors yields consistency across OMI and TROPOMI NO2 and HCHO columns within 10% (Section 4.4.4.)

**Table 1.** The slopes and offsets derived from various validations studies, used to bias correct the satellite retrievals used in the parameterization of PO3.

| Product                    | Slope | Offset                                       | Benchmark                           | Time period of validation                       | Reference              |
|----------------------------|-------|----------------------------------------------|-------------------------------------|-------------------------------------------------|------------------------|
| TROPOMI
NO 2 | 0.59  | 0.90×10 15 molecs/cm 2 | Global MAX-
DOAS
observations | 2018-2023                                       | Souri et al., (2025)   |
| TROPOMI
HCHO            | 0.66  | $0.32\times10^{15}$ molecs/cm 2   | Global FTIR observations            | 2018-2023                                       | Souri et al., (2025)   |
| OMI NO 2        | 0.83  | 0.26×10 15 molecs/cm 2 | Global MAX-DOAS observations        | Varies for each station spanning from 2010-2018 | Pinardi et al., (2020) |
| OMI
HCHO                | 0.79  | 0.82×10 15 molecs/cm 2 | Global FTIR observations            | Varies for each station spanning from 2004-2020 | et al.,                |

**3. Comparison with FNR**

The authors argue that the PO3 framework provides more detailed and continuous information on ozone production and sensitivity than the traditional formaldehydenitrogen ratio (FNR). I agree with this conceptual advantage. However, the manuscript does not clearly demonstrate how much additional information PO3 offers beyond FNR in a quantitative or diagnostic sense. It would strengthen the paper if the authors could illustrate specific cases or regions where PO3 reveals gradients or features that are not captured by FNR-based classifications.

The manuscript also describes FNRs as "binary," but it would be more accurate to say

that the interpretation of FNRs is often binary, based on thresholding between VOC-limited and NOx-limited regimes, while the FNR values themselves are continuous. Clarifying this distinction would help avoid oversimplifying the FNR framework.

Finally, the description of Figure 14 qualitatively compares the spatial patterns of PO3 sensitivities to HCHO and NO2. This section could be improved by quantifying those relationships, for example by providing correlation or regression metrics between PO3 sensitivities and FNRs, or by showing how the two indicators diverge under different chemical conditions (for example, high-HCHO/low-NO2 versus low-HCHO/high-NO2). Such quantitative comparisons would make the claimed improvement of the PO3 framework more convincing and scientifically interpretable.

**Response**

When someone gives us an FNR value (assuming no measurement errors and that HCHO and NO2 perfectly represent VOCR and reactive nitrogen), how do we actually assign a sensitivity value to it in units like ppbv/hr or 1/hr (in case of dPO3/dNO2 or dHCHO)? The main reason we use FNR is to help regulators. What regulators really need from us are the first and second-order derivatives of ozone production rates relative to NOX and VOC emissions, so they can estimate how PO3 will change under different emission scenarios. Where does FNR fit into this? It only tells us the sign of the sensitivities, and by itself it's not enough to translate varying FNR values into actual derivatives or sensitivities.

Going back to the original question: how do we provide these sensitivities given just an FNR value? That requires us to know a lot more about the underlying atmospheric conditions like light levels, humidity, how well HCHO and NO2 represent VOCR and reactive nitrogen, and in some cases heterogeneous chemistry, chlorine chemistry, HONO chemistry, and so on. FNR simply can't capture all these dimensions.

So, can our new product fully meet what regulators need? Not fully; but it is a step forward. It provides first-order derivatives based on how HCHO, NO2, water vapor, and photolysis rates change. We're missing second-order derivatives, and there's a more fundamental hurdle: HCHO and NO2 do not fully represent local emissions because they go through other physical and chemical processes like transport.

That said, we don't think it's very useful to compare an incomplete two-dimensional representation of multidimensional nonlinear ozone chemistry with a product that has more dimensions.

To elaborate more; FNR has three blind spots:

1. Lack of sensitivity magnitudes: FNR only classifies regimes without quantifying the actual magnitude of ozone sensitivities. For example, if  $\partial P_{O_3}/\partial NO_2$  is +10 s-1 or +3 s-1, both would be labeled "NOx-sensitive," even though their regulatory implications might be different. What truly matters about emission control is the magnitude of these responses. For this reason, CTM-based calculations (either

through direct decoupled methods, perturbation or adjoint approaches) are typically used. These, however, require extensive efforts to constrain model inputs with satellite data (see Souri et al., 2020:

https://acp.copernicus.org/articles/20/9837/2020/).

Our work provides quantitative first-order sensitivity maps, equivalent to directional derivatives (Appendix A), which is a major innovation of the new algorithm.

- 2. Varying FNR values cannot be directly linked to varying magnitude of sensitivities without accounting for photolysis rates, water vapor, and so forth. These geophysical variables are not articulated by FNR.
- 3. Lack of adequate dimensions: FNR slices the inherently multidimensional, nonlinear system into just two dimensions. To demonstrate this shortcoming, we perturbed photolysis rates over polluted regions during the KORUS-AQ campaign using observationally-constrained F0AM model. Multiplying photolysis rates by factors of 0.5 (dim, left), 1.0 (default, middle), and 2.0 (bright, right) produced three sets of PO3 isopleths.

The results clearly show that increasing light intensity raises both net PO3 and its sensitivities to NOx and VOC (the contours are more compact in the bright case; each contour corresponds to 3 ppbv/hr).

This means that the same FNR can correspond to entirely different magnitude of sensitivities depending on available light. There is where FNR falls apart.

Although one might expect FNR to indirectly reflect variations in photolysis rates, our analysis of 47,000 data points obtained from KORUS-AQ measurements showed no relationship between measured *j*NO2 and FNR:

A similar limitation arises from FNR's inability to account for water vapor effects on PO3. Capturing these complex nonlinear interactions between PO3, light, humidity, and precursor concentrations requires more advanced methods over a simple ratio, lacking any information about light intensity and humidity. In a data-driven framework, this is best achieved using nonlinear parameterizations such as DNNs.

This new product therefore represents a paradigm shift away from oversimplified FNR approaches. It not only provides spatiotemporal sensitivity magnitudes, but also accounts for multidimensional dependencies. We highlight this feature in Section 4.3.

**Modifications**

To better inform how the new sensitivity maps can eliminate the need for FNR, we added:

"Beyond Binary Maps from HCHO/NO2: A Deep Neural Network Approach to Global Daily Mapping of Net Ozone Production Rates and Sensitivities Constrained by Satellite Observations (2005–2023)"

While we had provided context about the advances made compared to FNR, we added a paragraph in the introduction describing why we should quantify the multidimensional magnitude of PO3 sensitivity, currently lacking in FNR-based approaches. We added in the introduction:

The overarching goal of producing ozone chemistry sensitivity maps is to inform regulatory agencies about the impact of emission reductions on locally produced ozone. Unlike conventional FNR-based binary maps, these maps must quantify the magnitude of sensitivity rather than merely indicating its direction. This quantitative approach is essential because both

the sign and magnitude of sensitivities are crucial for understanding the impact of emission changes. While detailed sensitivity maps can be derived from chemical transport models by perturbing underlying emissions, the lack of observational constraints on these models can introduce significant biases. Souri et al. (2025) attempted to address this limitation by providing magnitude-dependent sensitivity maps of PO3 to NO2 and HCHO using piecewise linear regression. However, their approach yielded derivatives of PO3 with respect to NO2 and HCHO that remained invariant with changes in light and humidity conditions. This limitation is problematic because reduced light conditions are known to substantially dampen the sensitivity of PO3 to NOx and VOCs, even under identical emission rates. The current work is therefore motivated by the need to capture the complex, multidimensional dependencies of PO3 on ozone precursors, light intensity, and humidity using a more flexible data-driven approach through a machine learning algorithm. While these maps will not replace process-based chemical transport model experiments, they can efficiently provide first-order assessments to: (i) strategize top-down modeling experiments, (ii) gauge the added value of satellites on predictions of PO3, and (iii) guide the design of sub-orbital missions in regions with poorly documented elevated PO3.

In the supplementary, we added a new section describing the fundamental issues with FNR; we did not include it in the main draft because it is more of a reminder for people who may misuse FNR rather than bringing new insights into ozone chemistry.

**1. FNR is oblivious to the impact of photolysis rates and water vapor content on $PO_3$**

The primary objective of using the formaldehyde-to-nitrogen dioxide ratio (FNR) is to reduce high-dimensional, non-linear ozone production rates into a two-dimensional framework based on volatile organic compound reactivity (VOCR) and reactive nitrogen. However, beyond the fact that HCHO and NO2 does not fully represent VOCR and reactive nitrogen, it is crucial to recognize that ozone production rate sensitivities and magnitudes depend on other geophysical variables independent of FNR. Among these variables, photolysis rates and water vapor are major drivers of atmospheric oxidation capacity, modulating numerous reactions related to ozone production (Kleinman et al., 2001).

To demonstrate photolysis rate effects on both PO3 magnitudes and sensitivities, we conducted F0AM box model simulations constrained by geophysical variables during June 6-9 of the KORUS-AQ campaign (Souri et al., 2025). We perturbed NOx, VOCs, and photolysis rates to generate three sets of isopleths (Figure S1). The results clearly show larger ozone production rates under more intense light conditions. More importantly, the contours corresponding to identical PO3 intervals (3 ppbv/hr) become more compact under brighter conditions, indicating that PO3 becomes more sensitive to both NOX and VOCs with increased light intensity. This pattern suggests that identical FNR values under different photolysis rates can have fundamentally different implications for ozone production rate sensitivities.

To confirm that FNR contains no photolysis rate information, we analyze paired FNR and jNO2 photolysis rate measurements from over 47,000 data points during the KORUS-AQ campaign, revealing no correlation between these variables (Figure S2). This demonstrates the need for additional dimensions in ozone sensitivity analysis, necessitating more sophisticated algorithms (like our approach) over traditional threshold-based methods.

**Figure S1.** The PO3 isopleths generated using F0AM box models derived from observations taken during the KORUS-AQ campaign under three different photolysis rates scenarios: (left) multiplied by 0.5, (middle) default, (right) multiplied by 2.0. Each contour represents 3 ppbv/hr.

**Figure S2.** The comparison of measured FNR and measured jNO2 frequencies taken from aircraft observations during the KORUS-AQ campaigns. All measured points are used to make this plot.

Figure S3 illustrates the representation of ozone sensitivities by mapping five variables derived from TROPOMI and our PO3DNN parameterization across two seasons over Los Angeles. FNR values are low during colder months due to abundant NO2 relative to HCHO, qualitatively suggesting the LA region should be predominantly VOC-sensitive. However, the derivatives and sensitivities of PO3 to both HCHO and NO2 remain muted due to limited photochemical activity, making PO3 unresponsive to NOX and VOC concentrations. Conversely, summer conditions yield larger derivatives, showing much stronger PO3 responses to both species. This example can be extended to different times of day, such as FNR values

from geostationary satellites or morning versus afternoon measurements from low Earth orbit satellites.

**Figure S3.** Five variables derived from our PO3DNN product based on TROPOMI dataset. The first row focuses on December-January-February (DJF), while the second row shows those variables for June-July-August 2023. The calculation of the sensitivities and derivatives are based on perturbation of the DNN algorithm described in the main paper.

The absence of PO3-relevant geophysical information in FNR also applies to water vapor. F0AM box simulations over polluted regions show that increasing humidity enhances PO3 through the generation of two OH molecules via H2O+O1D reactions (Figure S4). However, FNR contains no water vapor information, as humidity is driven by hydrological and meteorological factors decoupled from the processes determining FNR (Figure S5). This further necessitates adding water vapor as an additional dimension in ozone sensitivity

**Figure S4**. The effect of  $H_2O(v)$  on  $PO_3$  during KORUS-AQ campaigns. Only highly polluted regions (HCHO×NO2 > 10) are selected for this experiment.

**Figure S5.** The comparison of measured FNR and measured water vapor density taken from aircraft observations during the KORUS-AQ campaigns. All measured points are used to make this plot.

**4. Conversion factor and averaging kernel**

It is unclear whether satellite averaging kernels were applied when deriving the column-to-PBL conversion factors using MINDS. If they were applied, please specify how; if not, discuss the potential influence on near-surface concentrations and the resulting PO3 estimates.

**Response**

There are two main approaches to remove or mitigate the influence of the a priori assumptions used in OMI and TROPOMI AMFs in order to obtain a consistent, MINDS-driven conversion factor that reflects the satellite vertical sensitivity.

Approach 1: Convolving MINDS Conversion Factors with Satellite Averaging Kernels In this approach, the conversion factor is defined as

 $f_{AK} = q_{PBL}/\sum x$ ,

where  $q_{PBL}$  is MINDS PBL mixing ratio and  $x = x_a + A(x_{MINDS} - x_a)$ . Here,  $x_a$  and  $x_{MINDS}$  represent the a priori and MINDS partial columns, respectively, and A is the averaging kernels.

While this method is scientifically sound, it introduces significant complexity: the resulting conversion factor becomes dependent on satellite viewing geometry, scene-specific averaging kernels, and the a priori vertical profiles. This dependency makes validation of the conversion factors against in situ observations extremely difficult. As noted in the manuscript, the dominant source of systematic error in our product comes from the conversion factors themselves. If these factors are entangled with averaging kernel and a priori uncertainties, they lose generalization and consistency across retrievals and a priori frameworks. By maintaining a sensitivity- and a prioriagnostic formulation (as validated in Appendix B), we ensure that conversion factors can be robustly validated using aircraft observations and applied consistently across models. In other words, the question of "which model does better convert columns to the near surface concentrations?" can be more easily answered without delving into the nuances of satellite sensitivities.

Approach 2: Recalculating AMFs Using MINDS Vertical Shape Factors This alternative approach recalculates the AMFs using MINDS vertical profiles (section 2.1 in <a href="https://agupubs.onlinelibrary.wiley.com/doi/full/10.1002/2017JD028009">https://agupubs.onlinelibrary.wiley.com/doi/full/10.1002/2017JD028009</a>), allowing the conversion factor to remain independent of the satellite retrieval. This is a preferred algorithm over approach #1. However, it introduces a circular problem: recalculating AMFs would necessitate revalidating and bias-correcting TROPOMI and OMI NO2 and HCHO columns against ground-based datasets. Repeating the extensive work of Verhoelst et al. (2020), Vigouroux et al. (2021), Pinardi et al. (2021), and Ayazpour et al. (2025) would be a major undertaking.

For these reasons, we chose not to refine the TROPOMI and OMI VCDs using MINDS shape factors at the cost of introducing some biases in our product.

To show the impact of neglecting this step on PO3, we recalculate AMF with MINDS shape factors over the CONUS, an area with varying emissions and meteorological conditions, using our OI-SAT-GMI package (<a href="https://github.com/ahsouri/OI-SAT-GMI/tree/main">https://github.com/ahsouri/OI-SAT-GMI/tree/main</a>), and quantify the impact on PO3.

Regarding NO2, we see AMF (thus VCDs and PBL mixing ratios) to vary within 5±9% on average with minimal changes over polluted regions while seeing bigger values over higher latitudes or dark albedo where the retrieval becomes more dependent on the a priori. These changes induce minimal changes on PO3 (<20%) over PO3>0.5 ppbv area, especially hotpots of PO3. The changes can reach to 40-50% over remote high latitude regions, but PO3 errors are already extremely large (>200%) because of the errors in MINDS conversion factors:

Concerning HCHO, AMFs changed around the same magnitude as those of NO2 (10±7%) resulting in PO3 changing to <15% over PO3>0.5 ppbv/hr.

Therefore, skipping AMFs recalculation should result in ~25% errors in PO3 estimates. However, the consideration of AMFs without redoing the bias-correction would have resulted in the same level of errors, suggesting the most robust way is to adjust both (bias correction and AMFs) at the same time which is not feasible given our budget constraint.

**Modifications**

**We added:**

We also quantify the impact of inconsistent shape factors used in the retrievals and the MINDS profile on  $PO_3$  estimates and find them introducing systematic errors of 5-25% over  $PO_3>0.5$  ppbv/hr (Figures S14-S17). Refining TROPOMI and OMI products with MINDS shape factors would require reproducing several large-scale validation efforts (e.g., Verhoelst et al.,

2020; Vigouroux et al., 2021; Pinardi et al., 2021; Ayazpour et al., 2025), which is beyond the practical scope and resources of this study.

**In the summary section:**

The total errors budget emphasizes on the role of model used for converting satellite-based VCDs to near-surface concentrations and its importance for precisely determining ozone precursors levels near to the surface. Furthermore, in future efforts, we also need to refine satellite retrievals using spatially higher-resolution AMFs derived from MINDS while simultaneously performing retrieval validation against ground-based remote sensing observations.

iii) the inclusion of more sophisticated chemical mechanisms for the generation of the training dataset; and iv) enhanced representation of vertical profiles of NO2 and HCHO using observationally-constrained chemical transport models with more rigorous column to near-surface conversion factors (Cooper et al. 2020).

We added the above figures to the supplementary material.

**Minor Comments**

It would be helpful to clarify whether H2O values are directly inherited from MERRA-2 or modified within the MINDS model.

**Response**

MERRA2 is used to constrain U,V, QV, and T using the replay mode at 3-hourly basis in MINDS. So, meteorology is resolved in MINDS through GEOS. MERRA2 only adds a constraint.

**Modifications**

**We added:**

Meteorology is resolved using GEOS with several prognostic inputs, including water vapor, being constrained by MERRA-2 reanalysis using "replay" mode at 3-hourly basis (Orbe et al., 2017).

The description of "Southeast Asia" may be misleading; the text refers to August—September biomass burning, which applies mainly to maritime Southeast Asia, while continental Southeast Asia (Thailand, Myanmar, Laos, Cambodia) experiences its peak burning during February—April. Please clarify the regional definition.

**Response**

Thanks for pointing out this geographic mistake.

**Modifications**

We renamed the region to "maritime Southeast Asia" throughout the manuscript.

The expression "SZA acquired from the satellite L2 products" could be misleading, since SZA is not directly observed but computed from geometry information. Suggest rephrasing to "SZA derived from the geometry information in the L2 products."

**Response**

SZA is actually already computed and provided with L2 products. It's true that we can calculate that given time, location, and altitude, but the operation team has done it already.

**Modifications**

We modified it to:

Both SZA and surface altitude are provided as auxiliary fields in the satellite L2 products.

Check typographical errors (for example, "trend trends" to "trends"; "Tehan" to "Tehran").

**Response**

Corrected

The phrase "textbook example of non-linear chemistry" could be softened to "a clear demonstration of non-linear ozone chemistry."

**Response**

Corrected